# IMPROVING OPEN-VOCABULARY SEGMENTATION ACROSS DIVERSE DATA DISTRIBUTIONS

## ABSTRACT

Open-vocabulary segmentation (OVS) has gained attention for its ability to recognize a broader range of classes. However, OVS models show significant performance drops when applied to target data distributions beyond the source dataset. Fine-tuning these models on new datasets can improve performance, but often leads to the catastrophic forgetting of previously learned knowledge. To address this issue, we propose a method that allows OVS models to learn information from new data distributions while preserving prior knowledge. Our approach begins by evaluating the input sample's proximity to multiple data distributions, using precomputed multivariate normal distributions for each data distribution. Based on this prediction, we dynamically interpolate between the weights of the pre-trained decoder and the fine-tuned decoders. Extensive experiments demonstrate that this approach allows OVS models to adapt to new data distributions while maintaining performance on the source dataset.

## 1 INTRODUCTION

Open-vocabulary segmentation (OVS) has emerged as a pivotal area of research due to its potential to predict a diverse range of vocabularies without being restricted to a fixed set of predefined classes. This flexibility enables OVS models to identify new objects, rare categories, or arbitrary text-based descriptions. Recent advances in OVS (Xu et al., 2023; Yu et al., 2024) have extended its application to panoptic segmentation to recognize new classes across various segmentation tasks, such as semantic and instance segmentation.

Table 1: **Segmentation performance on Cityscapes and ADE20k.** We use Panoptic Quality (PQ) as the evaluation metric.

| Method | Vocab Type | Fine-tuning | Cityscapes | ADE20k |
|---|---|---|---|---|
| Mask2Former | Closed-set | ✓ | 62.1 | 39.7 |
| X-Decoder | OVS | ✗ | 36.2 | 16.7 |
| X-Decoder | | ✓ | **62.9** | **44.9** |
| fc-clip | OVS | ✗ | 44.0 | 26.8 |
| fc-clip | | ✓ | **64.2** | **47.6** |

Despite these advancements, we observe that OVS models are effective only within the data distribution of their source datasets. As shown in Table 1, OVS models perform significantly worse than closed-set segmentation models when evaluated on datasets with a different data distribution from the source dataset. Fine-tuning OVS models on these datasets can lead to substantial performance improvements; however, as illustrated in Figure 1, this comes at the cost of a significant drop in performance on unseen target data distributions. This limitation greatly restricts the applicability of OVS models in scenarios where recognizing objects in unseen target data distributions is critical.

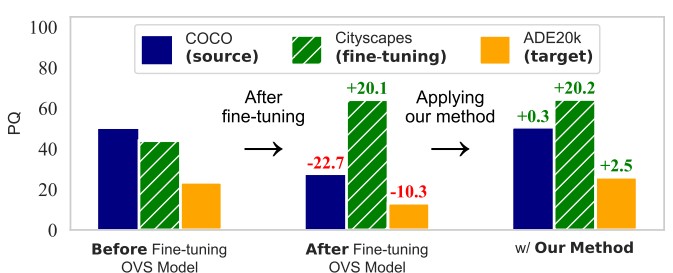

Figure 1: **Segmentation Performances before and after Applications of Fine-tuning and Our Method.** The numbers above bars indicate the relative performance gaps based on the original model (before fine-tuning). The fc-clip is used.

Continual learning methods offer a promising solution, as they learn new knowledge while preserving existing information. However, previous continual learning methods (Kirkpatrick et al., 2017; Kim et al., 2024) have limitations when applied to OVS models. We delve deeper into these challenges in Section 3.1.

We propose a new approach that enables OVS models to generalize to fine-tuning data distributions while preserving previous knowledge. This approach assumes that the fine-tuning dataset is already known, as it aims to improve the OVS model's performance on the fine-tuning dataset by training the model to align with its data distribution. Our method begins by training the decoder of the OVS model on the data distribution of the fine-tuning dataset. For this, we prepare a multivariate normal distribution (MVN) for each data distribution. During inference, we use these MVN distributions to infer interpolation factors that measure the proximity of the input sample to various data distributions. Based on this factor, we interpolate the weights of the pre-trained decoder and the fine-tuned decoders to generate new decoder weights for each input sample. This improves performance on the fine-tuning dataset while preserving performance on the source dataset, as shown in Figure 1. Our approach does not introduce additional parameters to the OVS model and integrates seamlessly with the existing OVS architecture.

In addition, we propose a novel evaluation protocol for OVS models that integrates methodologies from continual learning and OVS literature. This protocol considers all sequential training orders of COCO, Cityscapes, and ADE20K, and expands evaluations to include unseen datasets, such as DarkZurich, FoggyZurich, and GTA5, enabling a more comprehensive analysis.

Our experimental results demonstrate that applying the proposed approach to OVS models improves performance in the fine-tuning data distribution while maintaining performance in the previously seen data distribution. Specifically, when fine-tuned on Cityscapes (Cordts et al., 2016) and ADE20k (Zhou et al., 2019), the model adapts well to the fine-tuning data distribution without losing prior knowledge. We also observe the same effect when fine-tuning the model on multiple datasets. Furthermore, the performance improves on various target segmentation datasets, including Mapillary Vista (Neuhold et al., 2017), LVIS (Gupta et al., 2019), and BDD100k (Yu et al., 2020).

## 2 RELATED WORK

### 2.1 OPEN-VOCABULARY SEGMENTATION

Open-vocabulary segmentation (OVS) addresses the limitations of traditional closed-set segmentation models, which can only recognize predefined classes. Research on closed-set segmentation models has focused on identifying objects within a fixed set of classes. However, this restriction is impractical in real-world scenarios where it is crucial to recognize new or rare classes. OVS overcomes this issue by enabling the recognition of classes not included in the training.

Existing OVS literature mainly uses models trained on large external datasets to recognize novel classes. For example, Yu et al. (2024); Zhou et al. (2022); Ding et al. (2022); Wu et al. (2023) leverage CLIP (Radford et al., 2021), a large vision-language model, with OVS models to predict classes. Recent studies also explore methods such as using a pre-trained diffusion-based model (Xu et al., 2023) or combining the Segment Anything Model (SAM) (Kirillov et al., 2023) with CLIP to recognize a variety of classes (Yuan et al., 2024; Wang et al., 2024a). OVS models trained on large-scale datasets, such as X-Decoder (Zou et al., 2023a; 2024; 2023b), can handle OVS tasks as well as tasks like referring segmentation and image captioning. Despite these advancements, current OVS models, when not trained on specific datasets, can exhibit significantly lower performance. This paper addresses these unresolved issues in detail.

### 2.2 FINE-TUNING AND CATASTROPHIC FORGETTING

Fine-tuning is widely used to improve the performance of a pre-trained model on downstream tasks by adjusting the model's parameters (Yosinski et al., 2014; Kornblith et al., 2019). Recently, parameter-efficient fine-tuning (PEFT) has been introduced as an approach to effectively utilize the knowledge of pre-trained models. Instead of fine-tuning all parameters, PEFT adjusts only a subset to improve the performance of downstream tasks. These methods include linear probing, adapters (Houlsby

et al., 2019), low-rank adaptation (Hu et al., 2021), bias tuning (Cai et al., 2020), and visual prompt tuning (VPT) (Jia et al., 2022).

Although these methods improve task-specific performance, they often overlook the problem of catastrophic forgetting. Specifically, previous OVS fine-tuning methods primarily focus on adjusting the CLIP encoder to enhance segmentation performance, but they do not address catastrophic forgetting (Xu et al., 2024; Ghiasi et al., 2022; Li et al., 2022). We are the first to highlight and analyze this issue when fine-tuning OVS models on a new data distribution.

Many researchers have focused on replay-based continual learning methods to address catastrophic forgetting (Chaudhry et al., 2019; Shin et al., 2017). These methods help preserve previously acquired knowledge while the model learns new tasks by using past datasets. However, storing previous datasets can raise concerns about data storage, security, and privacy. To overcome these issues, exemplar-free continual learning methods, which do not store or use past datasets, have gained attention. In this area, parameter regularization methods (Kirkpatrick et al., 2017; Ritter et al., 2018; Liu et al., 2018), function regularization methods (Li & Hoiem, 2017; Dhar et al., 2019; Iscen et al., 2020), and architecture-based approaches are commonly used to solve the problem of catastrophic forgetting. Among these, architecture-based approaches include PEFT (Wang et al., 2022a; Liang & Li, 2024; Wang et al., 2022b; Smith et al., 2023), which introduces dedicated model parameters to facilitate learning new data.

Despite various efforts to address catastrophic forgetting in continual learning, this issue remains unresolved in OVS models. In this paper, we propose a novel method to overcome this problem and expand the range of data distributions that OVS models can recognize.

### 2.3 MULTI-SOURCE DOMAIN ADAPTATION

Multi-Source Domain Adaptation (MSDA) (Mansour et al., 2008) tackles the challenge of adapting models from multiple source domains to perform well on a single target domain. The primary focus of existing MSDA literature is the alignment of feature representations across multiple source domains and the target domain. For example, Li et al. (2021); Song et al. (2021); Peng et al. (2019) use multiple models from different source datasets to learn domain-specific representations to adapt knowledge from multiple sources to the target domain. In addition, Guo et al. (2018) introduce a mixture-of-experts approach for multi-source domain adaptation that explicitly models relationships between target examples and source domains.

The concept of using multiple models trained on diverse datasets in MSDA aligns with our approach. However, our method differs from MSDA in two key aspects: 1) addressing catastrophic forgetting in sequential learning scenarios, and 2) improving generalization not only to a single target data distribution but also to diverse distributions. This paper builds on these distinctions to develop a method that is better suited for open-vocabulary segmentation tasks in continual learning environments.

## 3 BACKGROUND

### 3.1 MOTIVATION

When fine-tuning the OVS model on the fine-tuning dataset, the model forgets previously learned knowledge. As shown in Figure 1, performance improves on the fine-tuning dataset after fine-tuning, but it significantly drops on the source dataset. To extend the data distributions that the OVS model can recognize, it is necessary to address this issue of catastrophic forgetting.

Whether the model is trained from scratch or fine-tuned on both the source and fine-tuning datasets, joint training consistently results in lower performance compared to training exclusively on the fine-tuning dataset. Notably, this holds true regardless of the distribution gap between the source and fine-tuning datasets, as training solely on the fine-tuning dataset yields better performance.

One common approach to preserving existing knowledge is joint training. In this method, the OVS model is trained simultaneously on the source and fine-tuning datasets, with each batch containing data from both datasets in equal proportions. This approach is inspired by previous studies that address balanced joint training across multiple datasets (Rolnick et al., 2019; Van de Ven et al., 2022) or multimodal datasets (Evans et al., 2024). However, this approach presents three issues: 1) Access to

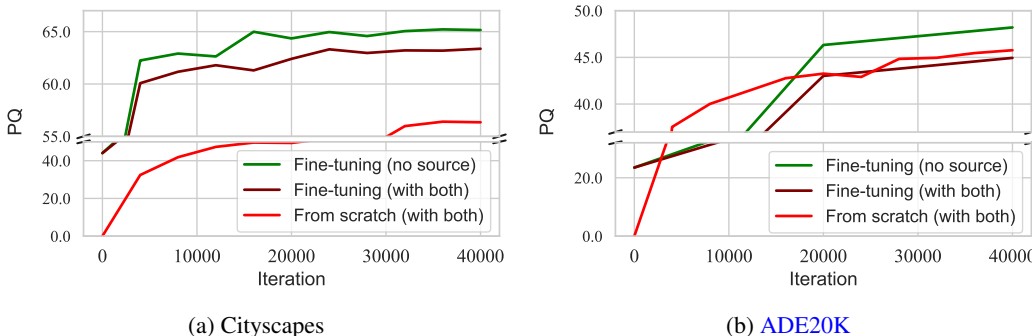

(a) Cityscapes                                   (b) ADE20K

Figure 2: **Comparison of performance trends for the OVS model fc-clip trained on both the source dataset and the fine-tuning dataset versus trained only on the fine-tuning dataset.** The left graph shows results when the fine-tuning dataset is Cityscapes, while the right graph corresponds to ADE20k. Evaluations are conducted on the validation set of the fine-tuning dataset.

all source datasets is required, which creates data management challenges. These challenges include issues with data usage rights, such as licensing. For instance, if a dataset's usage rights expire after it was used for training, joint training cannot proceed. 2) Whether the model is trained from scratch or fine-tuned on both datasets, joint training consistently results in lower performance compared to fine-tuning on the new dataset alone (see Figure 2). Specifically, this holds true regardless of whether the fine-tuning dataset is Cityscapes or ADE20K, as fine-tuning solely on the new dataset yields better performance. 3) In joint training, training datasets often contain different numbers of images. This difference can cause class imbalance, which hinders effective learning. Resolving this issue is a well-known challenge in the field (Johnson & Khoshgoftaar, 2019; Ghosh et al., 2024).

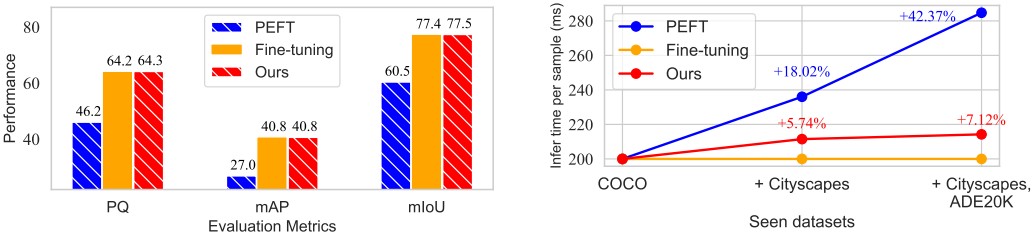

(a) Comparison of segmentation performance.          (b) Comparison of inference time.

Figure 3: (a) Segmentation performance comparison (`PQ`, `mAP`, `mIoU`) among standard fine-tuning, PEFT, and our method. All methods fine-tune fc-clip on the Cityscapes dataset. (b) Average inference time per sample compared across standard fine-tuning, PEFT, and our method, based on the number of datasets used during training. Average inference time per sample indicates the time required for a single sample to pass through the model during inference. The number of seen datasets includes the source dataset (COCO) and fine-tuning datasets (Cityscapes, ADE20k). All evaluations are conducted on the Cityscapes validation set.

Another approach is exemplar-free continual learning, which resolves data management issues by eliminating the need to store previous datasets. To explore this method, we apply visual prompt tuning (VPT) (Jia et al., 2022), a PEFT approach, to the OVS model. VPT has recently shown performance improvements in the field of continual learning (Qiao et al., 2023; Wang et al., 2022c). Following the method in Kim et al. (2024), we incorporate VPT into the OVS model by adding learnable prompts to the queries and positional embeddings of the model's decoder. However, applying this method to OVS models presents two challenges: 1) As shown in Figure 3a, PEFT results in lower performance on the new dataset compared to fine-tuning. This likely occurs because fine-tuning optimizes a larger set of parameters, leading to greater improvements (Wortsman et al., 2022). 2) As shown in Figure 3b, PEFT requires more inference time compared to our method and the baseline. While our method incurs increased inference time as the number of seen data distributions grows (Rypeść et al., 2024) due to the need to compute more interpolation factors for weight interpolation, it remains faster than techniques like PEFT that require additional parameters.

To address the limitations of previous techniques applied to OVS models, we propose a novel exemplar-free continual learning method. The proposed method starts by assessing the input sample's

proximity to multiple data distributions, using precomputed MVN distributions for each data distribution. Based on this, it dynamically interpolates the OVS model's decoder weights to generate decoder weights that suit the input sample. As shown in Figure 3, the proposed method improves performance on new datasets more effectively than PEFT, while using fewer computational resources.

**Problem Definition.** OVS models often struggle with desired unseen data distributions, limiting their applicability in real-world scenarios where new objects or classes frequently emerge. For instance, consider a scenario where a user deploys an OVS model in a driving scene. Pre-trained OVS models, without fine-tuning, perform poorly because they have not adapted to the driving scene's data distribution. On the other hand, fine-tuning the model on such data can compromise its open-vocabulary capabilities, restricting it to recognizing only objects and classes typical of the driving scene. This study addresses this issue by proposing a method that sequentially fine-tunes the model, extending its data distribution coverage while preserving its open-vocabulary properties.

More formally, we define our objective in detail as follows: The OVS model is first trained on the source dataset $D_{pr}^{\text{train}}$ and then fine-tuned sequentially on specific datasets $\{D_{ft,1}^{\text{train}}, D_{ft,2}^{\text{train}}, \dots\}$. Each dataset $D$ contains images $X_{img}$ and class labels $X_{text}$. At the $i$-th fine-tuning stage, the model only has access to the current training set $D_{ft,i}^{\text{train}}$ of the fine-tuning dataset. For evaluation, the model is evaluated on the test sets of source and fine-tuning datasets $\{D_{pr}^{\text{test}}, D_{ft,1}^{\text{test}}, \dots, D_{ft,i}^{\text{test}}\}$, and target datasets $\{D_{\text{target},1}, D_{\text{target},2}, \dots\}$. Note that the model has never encountered the target datasets during training.

# 4 METHODOLOGY

This section explains the proposed method that allows OVS models to learn a new data distribution without losing prior knowledge. First, it describes how to generate the MVN distributions for each data distribution during the training phase. Then, it provides a detailed explanation of the weight interpolation process in the inference phase. An overview of the inference process is shown in Figure 4.

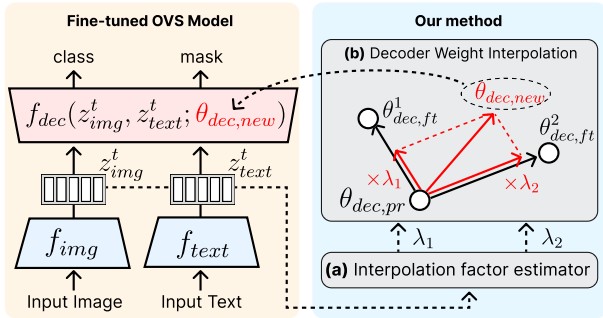

Figure 4: Inference process of our method.

## 4.1 TRAINING PHASE

During training, we first train the OVS model using the source dataset. Then, we fine-tune the trained OVS model on new datasets. Following the methods of Yu et al. (2024); Zou et al. (2023a), we keep the encoder fixed during fine-tuning and only update the decoder. Notably, our approach does not modify the original training process of the OVS model, including the objective function or architecture design.

Each time we train a dataset, we calculate two sets of means and covariance matrices from the image and text embeddings. These are components of the multivariate normal (MVN) distributions. After completing the training phase, we obtain the means and covariance matrices for the source dataset ($i = 0$) and the fine-tuning datasets ($i = 1, \dots, N_{ft}$), denoted as $\{\mu_{img}^i, \Sigma_{img}^i, \mu_{text}^i, \Sigma_{text}^i\}$.

## 4.2 INFERENCE PHASE

The inference process begins by calculating the interpolation factor vector $\boldsymbol{\lambda}$ (Algorithm 1, Steps 1-3). Specifically, the input image and text are passed through the encoder, producing embedding vectors for both. These embedding vectors are then fed into the probability density functions (pdf) of the image and text MVN distributions, which are defined for each data distribution. The image MVN distribution consists of $\mu_{img}^i$ and $\Sigma_{img}^i$, while the text MVN distribution consists of $\mu_{text}^i$ and $\Sigma_{text}^i$. This step produces a likelihood vector $l_{img} \in \mathbb{R}^{N_{ft}+1}$ for the image embedding and a likelihood vector $l_{text} \in \mathbb{R}^{N_{ft}+1}$ for the text embedding. A softmax function is then applied to these likelihood

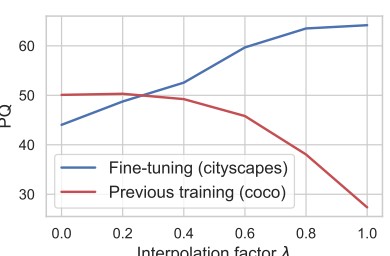

Figure 6: Illustration of $\boldsymbol{\lambda}$ generated by the interpolation factor estimator for input samples from seen and target data distributions.

vectors, resulting in $s_{img}$ and $s_{text}$. The interpolation factor $\lambda$ for each data distribution is determined by selecting the maximum value from both $s_{img}$ and $s_{text}$. By considering both the image and text, this approach calculates the appropriate interpolation factor for each data distribution. Section 5.1 demonstrates through an ablation study that using both image and text improves performance on new data distributions. Figure 4a shows the interpolation factor estimator that handles this process.

The calculated interpolation factor vector, $\boldsymbol{\lambda}$, is used to interpolate between the pre-trained decoder and the fine-tuned decoders (Ilharco et al., 2022) (Algorithm 1, Steps 4-5). Specifically, we multiply the difference between the weights of the distribution-specific fine-tuned decoder $\theta^i_{dec,ft}$ and the pre-trained decoder $\theta_{dec,pr}$ by the interpolation factor $\lambda_i$. This determines whether the final weights are closer to the pre-trained decoder or the fine-tuned decoder. After completing this process for all the fine-tuned decoders, we sum the results to form the final interpolated decoder. The weight interpolation process is illustrated in Figure 4b.

The decoder weight interpolation process determines whether the OVS model uses the weights fitted to the source dataset or the fine-tuning dataset, based on the interpolation factor. As shown in Figure 5, when $\lambda_i = 0$, the decoder uses the previously trained weights $\theta_{dec,pr}$, leading to strong performance on the source dataset. When $\lambda_i = 1$, the decoder applies the fine-tuned weights $\theta^i_{dec,ft}$, resulting in strong performance on the fine-tuning dataset. For $\lambda_i$ values between 0 and 1, the decoder interpolates between the two weights, achieving moderate performance on both datasets.

Finally, the resulting decoder weights are used to predict the mask and class for the embedding of the input. The complete inference procedure with interpolation of the decoder weights, is outlined in Algorithm 1.

Figure 5: Performance on the validation set of Cityscapes and COCO depending on the interpolation factor $\lambda$, using fc-clip.

**Discussion.** We observe that our method behaves differently depending on whether the input sample is close to the seen data distribution or the target data distribution. Figure 6 shows an example of the $\boldsymbol{\lambda}$ produced by the interpolation factor estimator. When the input sample is from the seen data distribution, it generates values close to 0 or 1. This indicates that a distribution-specific model is selected for the input sample. This behavior is effective because using the model trained on the corresponding data distribution is optimal when the input sample is close to the seen data distribution.

On the other hand, for samples from the target data distribution, the interpolation factors are more evenly distributed between 0 and 1. This means that our method combines the models trained on seen data distributions to prevent the model from relying on a single data distribution. As a result, this approach improves generalization performance on input samples from the target data distribution. We demonstrate this in the Section 5.

## 5 EXPERIMENTS

**Settings.** For panoptic segmentation, fc-clip and X-Decoder use COCO as the pretraining dataset and are fine-tuned on Cityscapes and ADE20k. We evaluate both models on eight unseen datasets using task-specific metrics (mIoU, PQ, mAP), reporting PQ in the main paper and including others

Table 2: Performance comparison between standard fine-tuning, previous continual learning methods, and our method, with COCO as the source dataset. All methods fine-tune the models using (a) Cityscapes or (b) ADE20K datasets. `PQ` is used.

**(a) Cityscapes**

| Method | COCO (source) | Cityscapes (fine-tuning) | ADE20K (target) | Avg on 6 datasets |
|---|---|---|---|---|
| fc-clip | 50.1 | 44.0 | 23.5 | 45.6 |
| + Fine-tuning | -22.7 | +20.1 | -10.3 | -3.9 |
| + Joint training | +0.6 | +17.9 | +1.7 | +0.5 |
| + ER | -1.6 | +19.0 | +0.3 | -0.6 |
| + LwF | -10.7 | +12.2 | -0.8 | -1.1 |
| + EWC | -25.9 | +19.3 | -9.8 | -4.3 |
| + ECLIPSE | -6.0 | +2.2 | +0.9 | -0.7 |
| **+ Ours** | **+0.3** | **+20.2** | **+2.5** | **+0.6** |
| X-Decoder | 56.7 | 36.3 | 16.7 | - |
| + Fine-tuning | -50.4 | +26.6 | -12.9 | - |
| **+ Ours** | **-0.4** | **+26.6** | **+0.1** | - |

**(b) ADE20K**

| Method | COCO (source) | ADE20K (fine-tuning) | Cityscapes (target) | Avg on 6 datasets |
|---|---|---|---|---|
| fc-clip | 50.1 | 23.5 | 44.0 | 46.0 |
| + Fine-tuning | -7.7 | **+24.1** | -3.0 | +0.3 |
| + Joint training | +1.4 | +16.5 | -1.2 | +0.6 |
| + ER | +0.4 | +21.5 | -3.5 | +0.0 |
| + LwF | -3.8 | +13.7 | -1.0 | -0.4 |
| + EWC | -11.1 | +20.7 | -2.6 | -1.5 |
| + ECLIPSE | -0.5 | +0.2 | -5.9 | -0.4 |
| **+ Ours** | **+1.7** | +23.8 | **-0.3** | **+0.6** |
| X-Decoder | 56.7 | 16.7 | 36.3 | - |
| + Fine-tuning | -37.3 | +28.2 | -3.7 | - |
| **+ Ours** | **-1.5** | **+29.2** | **+1.4** | - |

Table 3: Performance comparison among standard fine-tuning, previous continual learning methods, and our method, with ADE20K as the source dataset. All methods fine-tune the models using (a) COCO or (b) Cityscapes datasets. `PQ` is used.

**(a) COCO**

| Method | ADE20K (source) | COCO (fine-tuning) | Cityscapes (target) |
|---|---|---|---|
| fc-clip | 48.1 | 42.3 | 40.9 |
| + Fine-tuning | -18.5 | **+10.4** | +3.3 |
| **+ Ours** | **-1.3** | +9.3 | **+5.2** |

**(b) Cityscapes**

| Method | ADE20K (source) | Cityscapes (fine-tuning) | COCO (target) |
|---|---|---|---|
| fc-clip | 48.1 | 40.9 | 42.3 |
| + Fine-tuning | -18.5 | **+21.4** | -11.5 |
| **+ Ours** | **+0.0** | +19.5 | **+0.0** |

in the appendix. During fine-tuning, we freeze the encoders and train the decoders, implementing an interpolation factor estimator with a softmax temperature of 0.01 and log-likelihoods of MVN distributions. Detailed descriptions of datasets, evaluation metrics, and implementation details are provided in the appendix.

## 5.1 COMPARISON WITH OTHER METHODS

In each experiment, we evaluate the model on the source dataset, the fine-tuning dataset, and the target dataset. When the model is fine-tuned on Cityscapes, we treat ADE20K as the target dataset for evaluation, and vice versa.

**Results of fine-tuning with Cityscapes.** We present the evaluation results in Table 2a after fine-tuning the model on Cityscapes. Our method improves performance on the fine-tuning dataset while maintaining the performance on the source dataset, regardless of whether it is applied to fc-clip or X-Decoder. Specifically, compared to fine-tuning, our method preserves performance on the source dataset more effectively (e.g., Fine-tuning: $-22.7$, Ours: $+0.3$ for fc-clip / Fine-tuning: $-50.4$, Ours: $-0.4$ for X-Decoder), while achieving the same improvements on the fine-tuning dataset. In addition, we observe that the performance improvement of joint training is relatively smaller compared to our method (e.g., Joint training: $+17.9$, Ours: $+20.2$ for fc-clip). Furthermore, we observe that other continual learning methods consistently result in performance degradation on the source dataset (e.g., ER: $-1.6$, LwF: $-10.7$, EWC: $-25.9$, ECLIPSE: $-6.0$). In contrast, our method preserves performance on the source dataset (e.g., Ours: $+0.3$) and achieves better results on both the fine-tuning and target datasets.

**Results of fine-tuning with ADE20K.** We present the evaluation results of our method and previous methods when fine-tuning on ADE20K in Table 2b. Since ADE20K shares a similar data distribution with COCO, previous methods maintain performance on the source dataset compared to fine-tuning on Cityscapes. However, they still show a consistent performance drop on target datasets. In contrast, our method improves performance on the target dataset while also enhancing results on the source dataset and achieving a significant boost on the fine-tuning dataset (e.g., fc-clip with ours: source $+1.7$, fine-tuning $+23.8$, target $-0.3$). The improvement on the source dataset indicates that our method not only preserves prior knowledge but also enhances performance in the previously trained

data distribution by leveraging new knowledge. Additionally, X-Decoder loses performance on the source dataset with standard fine-tuning, but with our method, this performance is effectively preserved (e.g., X-Decoder with ours: $-1.5$ on the source dataset).

**Results with ADE20K as the source dataset.** To evaluate whether the proposed method shows superior performance when using ADE20K as the source dataset instead of COCO, we conduct additional experiments. As shown in Table 3, the proposed method preserves the performance of the source dataset while improving the performance on the fine-tuning dataset. It achieves consistent performance improvements on target datasets that are not included during training (e.g. Ours: $+5.2$ for Cityscapes, $+0.0$ for COCO).

**Results of fine-tuning with multiple datasets.** As shown in Table 4, we compare the standard fine-tuning method with our approach in the sequential training scenario on ADE20K and Cityscapes. Fine-tuning results in a significant performance drop on source datasets (e.g., ADE→City: $-29.3$, City→ADE: $-10.8$ on COCO), maintaining strong performance only on the most recent training dataset. In contrast, our

Table 4: Performance of standard fine-tuning and our proposed method. The best performance for each dataset is underlined. City→ADE refers to the model fine-tuned on the Cityscapes dataset first, followed by ADE20K. The reverse applies to ADE→City. `PQ` is used.

| Method | The order of fine-tuning | COCO (source) | ADE20K (fine-tuning 1) | Cityscapes (fine-tuning 2) |
|---|---|---|---|---|
| fc-clip | - | 50.1 | 23.5 | 44.0 |
| + Fine-tuning | ADE → City | 20.8 | 15.4 | 65.2 |
| + Fine-tuning | City → ADE | 39.3 | 48.3 | 46.0 |
| + Joint training | City, ADE | 48.6 | 35.5 | 60.5 |
| + Ours | City, ADE | **51.6** | **47.0** | **64.3** |

method improves performance on the source dataset (e.g., $+1.5$ on COCO) and enhances results across all fine-tuning datasets. Furthermore, joint training achieves high performance on the source dataset compared to sequential fine-tuning but performs worse than our method across all three datasets.

Table 5: Performance comparison between sequential training and our method on 8 unseen datasets. `PQ` is used.

| Method | Source Dataset | The order of fine-tuning | LVIS (mAP) | BDD100K (PQ) | Mapillary (mIoU) | PC-59 (mIoU) | PC-459 (mIoU) | PAS-20 (mIoU) | PAS-21 (mIoU) | A-847 (mIoU) |
|---|---|---|---|---|---|---|---|---|---|---|
| OpenSeeD | COCO,Object365 | - | 14.4 | 10.7 | 15.0 | 47.7 | 11.0 | 87.2 | 33.5 | 5.3 |
| fc-clip | COCO | - | 20.5 | 19.0 | 26.0 | 53.0 | 16.9 | 93.1 | 80.2 | 13.8 |
| + Fine-tuning | COCO | City → ADE | 21.7 | 19.7 | 27.8 | 52.1 | 17.2 | 92.3 | 76.7 | 16.0 |
| + Fine-tuning | COCO | ADE → City | 10.4 | 21.3 | 24.2 | 45.9 | 13.5 | 87.4 | 70.7 | 11.5 |
| + Joint training | COCO,City,ADE | - | 10.4 | 21.3 | 24.2 | 45.9 | 13.5 | 87.4 | 70.7 | 11.5 |
| + Ours | COCO | City, ADE | **23.1** | **22.6** | **29.1** | **54.9** | **17.9** | **93.6** | **80.7** | **16.3** |

As shown in Table 5, we compare the fine-tuning technique with our method and the previous OVS model, OpenSeeD (Zhang et al., 2023), on target datasets. We observe that OpenSeeD performs worse than fc-clip, which is trained solely on COCO, across the eight target datasets. Fine-tuning fails to consistently improve performance on these datasets, and in some cases, it even results in performance drops (e.g., City→ADE: $-3.3$ on PAS-21, ADE→City: $-11.1$ on LVIS). In contrast, our method achieves consistent performance improvements across all target datasets. In addition, joint training shows better generalizability than sequential fine-tuning but still underperforms compared to our method.

## 5.2 METHOD ANALYSIS & ABLATION STUDY

**Analysis on Seen and Truly Unseen Classes.** This section analyzes the performance of our method on seen and truly unseen classes. We use COCO as the source dataset, Cityscapes for fine-tuning, and ADE20K for evaluation. Truly unseen classes refer to those not present in either COCO or Cityscapes. Seen classes include those present

Table 6: Comparison of performance on seen and truly unseen classes. `mIoU` is used.

| Method | Seen Classes | Truly Unseen Classes |
|---|---|---|
| fc-clip | 35.0 | 28.6 |
| + Ours | **37.9** | **30.9** |

in at least one of these datasets. Our method achieves performance improvements for both seen classes (Ours: 37.9, fc-clip: 35.0) and truly unseen classes (Ours: 30.9, fc-clip: 28.6) compared to the original fc-clip, as shown in Table 6. This suggests that merging the domain-specific knowledge of the two OVS model decoders through our weight interpolation technique truly enhances the generalization capability for target datasets.

**Evaluation on Diverse and Challenging Domains.** We evaluate our method on datasets that differ significantly from the training dataset's domain to demonstrate its robustness. The evaluation includes GTA5, a synthetic driving dataset, and DarkZurich and FoggyZurich, which consist of nighttime and foggy driving scenes. These datasets introduce substantial domain shifts compared to ADE20K and COCO, which are used as the training and fine-tuning datasets, respectively.

As shown in Table 7, the results show that standard fine-tuning of fc-clip reduces performance across all three datasets. In contrast, our interpolation-based method improves performance by leveraging both the original and fine-tuned parameters. This demonstrates that our approach effectively handles target domains with large domain differences, including adverse conditions and synthetic environments.

Table 7: Performance comparison (`mIoU`) on datasets with significant domain shifts.

| Method | GTA5 | DarkZurich | FoggyZurich |
|---|---|---|---|
| fc-clip | 65.6 | 40.2 | 54.4 |
| + Fine-tuning | 58.4 | 39.8 | 52.1 |
| + Ours | **66.6** | **43.1** | **55.9** |

**Ablation Study of Image and Text Distribution.** In our method, we determine the interpolation factor using the MVN distributions of both image and text data. We conduct an analysis by removing either the image or text distribution and comparing the results to the case where both distributions are used (Table 8). The best performance is observed when both image and text distributions are used, as this combination not only improves performance on the fine-tuning dataset but also ensures stability on target datasets. This result shows that combining these distributions allows for more accurate selection of interpolation factors for the fine-tuning dataset.

Table 8: Comparison between using both image and text or using only one type of information. Fine-tuned fc-clip on Cityscapes. Unseen represents the average score across 8 target datasets. `PQ` is used.

| Distribution | COCO (source) | Cityscapes (fine-tuning) | Unseen |
|---|---|---|---|
| image only | 51.5 | 43.4 | 40.3 |
| text only | **51.9** | 60.7 | 40.6 |
| image + text | 51.6 | **64.3** | **40.9** |

**Comparison of Alternative Prototype Models with the MVN Distribution.** Table 9 presents the evaluation results comparing three different prototype models available for estimating interpolation factors in our method. K-means clustering causes significant performance loss on the source dataset, and kernel density estimation fails to improve performance on the fine-tuning dataset. In contrast, the MVN distribution not only maintains performance on the source dataset and improves performance on the fine-tuning dataset but also achieves consistent results on target datasets. These findings emphasize the versatility of the MVN distribution in adapting to various datasets.

Using only the MVN distribution poses challenges in capturing the data distribution of samples because our algorithm does not involve clustering. However, the MVN distribution still performs well. This is because a small distribution gap between datasets, where the two domains become indistinguishable, often indicates that the datasets share similar distributions. In such cases, OVS models are expected to perform well, requiring minimal reliance on our algorithm.

Table 9: Analysis of the prototype modeling in the interpolation factor estimator. We fine-tune fc-clip on Cityscapes. Unseen represents the average score across 8 target datasets. `PQ` is used.

| Prototype Models | COCO (source) | Cityscapes (fine-tuning) | Unseen |
|---|---|---|---|
| k-means clustering | 42.4 | 64.1 | 40.6 |
| kernel density estimation | 48.1 | 57.4 | 40.6 |
| MVN distribution | **50.4** | **64.3** | **40.9** |

**Replacing Weight Interpolation with Prompts.** In this experiment, we compare the performance of replacing our method's weight interpolation (Algorithm 1, Steps 4-5) with prompt-based alternatives. The prompt implementation follows these steps: 1) For each data distribution, we train only the decoder's query and positional embeddings, then store them in a prompt pool. 2) During inference, we compute interpolation factors for each data distribution using our method. 3) We select the data distribution with the highest interpolation factor and replace the original decoder's query and positional embeddings with those from the corresponding prompt in the prompt pool (Wang et al., 2022a). As shown in Table 10, the prompt-based approach results in lower performance compared to our method on both the source and the fine-tuning dataset. Additionally,

Table 10: Comparison between the prompt-based approach and our weight interpolation. We fine-tune fc-clip on Cityscapes. Unseen represents the average score across 8 target datasets. `PQ` is used.

| Method | COCO (source) | Cityscapes (fine-tuning) | Unseen |
|---|---|---|---|
| Prompt | 43.3 | 48.9 | 39.1 |
| Weight interpolation | **50.4** | **64.3** | **40.9** |

our method outperforms the prompt-based approach on target datasets. Therefore, we conclude that weight interpolation is more effective for our task than using the prompt-based approach.

## 5.3 COMPUTATIONAL RESOURCES

Table 11: Inference time per sample with varying numbers of seen datasets. The unit for all numbers in the table is milliseconds (ms).

| Number of Seen Datasets | Encoder | Interpolation Factor Estimator | Decoder Weight Interpolation | Decoder | Total Inference Time Per Sample | Change (%) |
|---|---|---|---|---|---|---|
| 1 | 97.69 | - | - | 102.30 | 199.99 | +0.00% |
| 2 | 97.69 | 0.81 | 10.69 | 102.30 | 211.48 | +5.75% |
| 3 | 97.69 | 1.01 | 13.23 | 102.30 | 214.23 | +7.12% |

Our method ensures efficient use of computational resources during inference. It avoids the additional parameters required by other continual learning techniques as the number of learned datasets grows (Kim et al., 2024). Furthermore, our method does not involve multiple forward passes (Nicolas et al., 2023; Wang et al., 2022a), which are computationally expensive. Instead, we perform weight interpolation exclusively in the decoder of encoder-decoder models, minimizing overhead.

To demonstrate the efficiency of our method, we measure inference time as the total processing time per sample, as shown in Table 11. The increase in inference time remains minimal as the number of datasets grows. Specifically, training with two datasets increases inference time by only $5.75\%$p, while adding a third dataset results in a marginal additional increase of $1.37\%$p. These results confirm the scalability of our approach with respect to inference time.

In addition to computational efficiency, our method achieves significant storage savings. Unlike ensemble-based approaches, which require storing the entire model for each dataset (Wortsman et al., 2022; Khirodkar et al., 2022), our method stores only the decoder parameters. This reduces the storage requirement to $6.11\%$ of the total model size, which corresponds to approximately 80MB per dataset. This efficiency ensures scalability in scenarios involving multiple datasets.

## 6 LIMITATIONS

Our method incurs computational overhead during the weight interpolation process, as illustrated in Table 11. This presents a significant challenge, as it reduces the efficiency of OVS models, and remains an unresolved issue insufficiently addressed in prior research. To address this problem, reducing the number of parameters involved in interpolation could be a potential solution. This can be achieved by exploring approaches from prior work on model merging, such as pruning techniques (Yadav et al., 2024; Sun et al., 2023), which eliminate redundant parameters, or Mixture-of-Experts methods (Tang et al., 2024), which activate only a subset of parameters for specific tasks.

However, applying these techniques to segmentation models, particularly OVS models, introduces unique challenges due to their structural characteristics and the complexity of the data. Developing methods to reduce the cost of weight interpolation is a critical research direction that can overcome these limitations and optimize the inference time of OVS models.

## 7 CONCLUSION

Conventional segmentation models are limited to recognizing predefined classes, which highlights the growing importance of Open-vocabulary Segmentation (OVS) for broader category prediction. However, OVS models show reduced performance when applied to target datasets beyond the source dataset. While fine-tuning OVS models improves performance on fine-tuning datasets, we observe that it leads to catastrophic forgetting of previous knowledge. To address this issue, we propose a method that adaptively interpolates between the weights of the pre-trained decoder and the fine-tuned decoders based on the input sample's proximity to different data distributions. We conduct extensive experiments to verify the method, showing that it allows OVS models to effectively learn on fine-tuning data distributions while preserving prior knowledge.

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

APPENDIX

## A OPEN-VOCABULARY SEGMENTATION

We define the input image and class label as $x_{img}$ and $x_{text}$, respectively. The image encoder and text encoder are defined as $f_{img}$ and $f_{text}$, with parameters $\theta_{img}$ and $\theta_{text}$ representing the parameters of the image and text encoders. The image embedding is computed as $z_{img} = f_{img}(x_{img}; \theta_{img})$, and the text embedding is computed as $z_{text} = f_{text}(x_{text}; \theta_{text})$. The decoder takes these embeddings as input and predicts $N_q$ pairs of masks and class labels, where $N_q$ is the number of object queries in the decoder. Specifically, the decoder, $f_{dec}$, takes $z_{img}$ and $z_{text}$ as inputs and predicts the output $o = f_{dec}(z_{img}, z_{text}; \theta_{dec})$. The output $o$ consists of $N_q$ pairs of masks and class embeddings, $\{(m_i, c_i)\}_{i=1}^{N_q}$, where $i$ denotes the index of the pair, $m_i$ represents the mask, and $c_i$ represents the corresponding class embedding. The class associated with mask $m_i$ is determined by selecting the class label with the highest similarity between the predicted class embedding $c_i$ and the text embedding $z_{text}$. This approach allows the model to predict a wide range of classes without being limited to predefined categories.

## B EXPERIMENT SETTINGS

**Datasets.** For the panoptic segmentation task, fc-clip and X-Decoder use COCO (Lin et al., 2014) as the source dataset. For the fine-tuning datasets, we use Cityscapes (Cordts et al., 2016) and ADE20k (Zhou et al., 2019). For evaluation purposes only, we assess model performance on eight target datasets: i) LVIS (Gupta et al., 2019), ii) BDD100K (Yu et al., 2020), iii) Mapillary Vista (Neuhold et al., 2017), iv) Pascal Context (Mottaghi et al., 2014) with 59 common classes (PC-59), v) Pascal Context with all 459 classes (PC-459), vi) PASCAL VOC (Everingham et al., 2010) with 20 foreground classes (PAS-20), vii) an extension of PAS-20 with an additional background class (PAS-21), and viii) A-847, which includes all 847 classes from ADE20K (Zhou et al., 2019).

**Evaluation Metrics.** We evaluate all OVS models on the tasks of open-vocabulary panoptic, instance, and semantic segmentation. For evaluation, we use the Panoptic Quality (PQ) (Kirillov et al., 2019), mean Average Precision (mAP), and mean Intersection over Union (mIoU) metrics. When evaluating on eight different unseen datasets, we select the most representative metric for each dataset based on the task it targets. Specifically, mIoU is used for semantic segmentation tasks, PQ for panoptic segmentation, and mAP for instance segmentation. In our experiments, PQ, mAP, and mIoU show similar performance trends. To maintain clarity, we only report PQ in the main paper and include the other metrics in the appendix.

**Implementation Details.** We apply our method to two OVS models: fc-clip (Yu et al., 2024) with ConvNext-L (Liu et al., 2022) backbone and X-Decoder (Zou et al., 2023a) with Focal-L (Yang et al., 2022) backbone. The fc-clip uses the CLIP (Radford et al., 2021) for both the image and text encoders, and training only decoder of the model using COCO (Lin et al., 2014). X-Decoder trains its encoder and decoder on the multiple pre-training datasets, including COCO, SBU Captions (Ordonez et al., 2011), Visual Genome (Krishna et al., 2017). Following the fc-clip and X-Decoder, we freeze the encoders and train only the decoder for both OVS models during fine-tuning. To implement the interpolation factor estimator in our method, we use the softmax temperature $T$ as $0.01$ for the softmax operation, and calculate the log-likelihood for the MVN distribution.

## C COMPARED METHODS

Since there is no prior research that apply continual learning to OVS models, we apply the previous continual learning methods to the OVS models and evaluate all approaches. Following Wang et al. (2024b); Chen & Liu (2022); Parisi et al. (2019); Mundt et al. (2023), we categorize previous methods into replay-based, regularization-based (parameter, function), and architecture-based approaches. We apply a representative method from each category to OVS models and compare their performance.

**Replay-based Method.** Experience Replay (ER) serves as the conceptual foundation for many memory-based methods (Lopez-Paz & Ranzato, 2017; Iscen et al., 2020). In our experiments, we apply this technique to the OVS model. ER stores a subset of training samples from previous datasets

and uses them during the training of a new dataset. For ER, we select 10 training samples per class from the source dataset. Unlike our method, ER requires access to the source dataset during the training of a new dataset, which makes a fair comparison difficult.

**Function Regularization.** We incorporate Learning without Forgetting (LwF) (Li & Hoiem, 2017), a function regularization method, into the OVS model. LwF is an exemplar-free continual learning method that uses knowledge distillation loss based on the distance between predictions of the pre-trained model and the fine-tuned model. This loss helps regularize the model to preserve its prior knowledge.

**Parameter Regularization.** The Elastic Weight Consolidation (EWC) (Kirkpatrick et al., 2017) method is adapted for the OVS model. EWC is a parameter regularization approach that does not rely on previous datasets. It first estimates the importance of each neuron by calculating the Fisher information matrix. This matrix assigns weights to the distance between the parameters of the pre-trained model and the fine-tuned model. This process suppresses changes to parameters that are crucial for preserving previous knowledge.

**Architecture-based Method.** We apply ECLIPSE (Kim et al., 2024), one of the architecture-based methods, to the OVS model. This method is designed for class-incremental learning in closed-set segmentation tasks and does not rely on the previous dataset. ECLIPSE introduces visual prompt tuning for the decoder by adding learnable prompts to the object queries and positional embedding. For our task, we add 250 prompts for each fine-tuning data distribution to ensure sufficient learning capacity. We use only the prompt tuning component of ECLIPSE in the OVS model and do not include the classifier or logit manipulation components.

# D DISCUSSION & ANALYSIS

Table A1: Performance comparison between the argmax and softmax operations in the interpolation factor estimator. We use fc-clip with our method and fine-tune it on both Cityscapes and ADE20K. `PQ` is used.

| Decision Rule | Fine-tuning Dataset | LVIS (mAP) | BDD100K (PQ) | Mapillary (mIoU) | PC-59 (mIoU) | PC-459 (mIoU) | PAS-20 (mIoU) | PAS-21 (mIoU) | A-847 (mIoU) |
|---|---|---|---|---|---|---|---|---|---|
| Argmax | Cityscapes, ADE20k | 21.3 | 18.3 | 26.9 | 53.1 | 17.0 | 93.2 | 80.2 | **16.3** |
| Softmax | Cityscapes, ADE20k | **23.1** | **22.6** | **29.1** | **54.9** | **17.9** | **93.6** | **80.7** | 16.3 |

**Replacing Softmax with Argmax.** In this study, we use the softmax function to calculate interpolation factors for each data distribution. Considering that argmax is a hard version of softmax, we compare the segmentation performance on target datasets when using argmax and softmax operations. Table A1 presents the evaluation results. We observe that softmax consistently outperforms argmax across all target data distributions (e.g., on LVIS, argmax: 21.3, softmax: 23.1). In the PAS-20, PAS-21, and A-847, there is little difference in performance between softmax and argmax. This is because the interpolation factor from softmax tends to be close to 0 or 1 when the input sample is close to the seen data distribution, making softmax behave similarly to argmax. As shown in Figure D1, for A-847, the interpolation factors are close to 0 or 1 because it shares a data distribution similar to ADE20K, a training dataset. In contrast, the interpolation factors for BDD100K are evenly distributed between 0 and 1. This occurs because BDD100K is closer to an target data distribution. In this case, our method improves generalization performance by combining models trained on multiple data distributions. These results indicate that considering multiple data distributions simultaneously via softmax leads to better performance than selecting a single data distribution through argmax, supporting the effectiveness of our design choice.

**Extending the Proposed Method to Traditional Continual Learning.** Our approach can also be extended to traditional continual learning tasks. In this context, recent techniques such as prompt-tuning (Wang et al., 2022a) and LoRA (Liang & Li, 2024) maintain independent parameter sets for each incremental session, enabling task-specific adaptation. Similarly, our method leverages independent parameter sets generated during each incremental session and uses the initial model to estimate the data distribution proximity of the input sample. This allows the method to dynamically merge the corresponding parameters, enabling accurate predictions for traditional continual learning tasks while effectively mitigating catastrophic forgetting. This adaptability demonstrates the broader potential of our framework beyond OVS task.

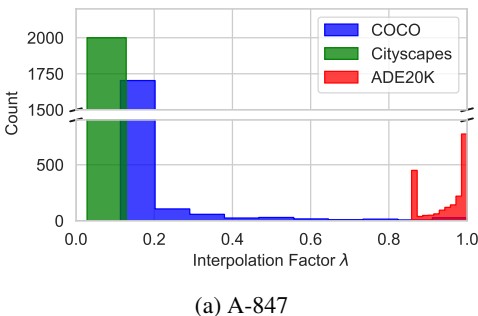 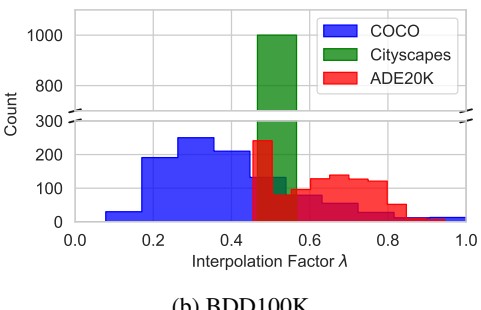

(a) A-847          (b) BDD100K

Figure D1: The histogram of interpolation factors when inferring all samples from the validation sets of (a) A-847 and (b) BDD100K. We fine-tune fc-clip on Cityscapes and ADE20K and use `PQ` as the evaluation metric.

**Hyperparameter Sensitivity Analysis.** We analyze the impact of the softmax temperature $T$ used to compute interpolation factors in our method. While our approach introduces no additional hyperparameters related to the MVN distributions, the softmax temperature critically influences the effectiveness of interpolation. Table A2 presents the results of our ablation study.

Table A2: Effect of softmax temperature $T$ on performance across datasets. Results are reported as mIoU.

| $T$ | Previous (COCO) | Fine-tuning (ADE20K) | Unseen (Cityscapes) | Total |
|---|---|---|---|---|
| 0.0001 | 50.7 | 35.4 | 43.8 | 129.9 |
| 0.001 | 51.2 | 42.2 | **43.9** | 137.3 |
| 0.01 | **51.8** | 47.3 | 43.7 | **142.8** |
| 0.1 | 51.3 | **47.5** | 43.2 | 142.0 |
| 1.0 | 51.2 | 47.4 | 43.2 | 141.8 |

We observe that using a small temperature T reduces performance on the fine-tuning dataset due to excessive smoothing of the interpolation factors. This results in minimal contribution from the fine-tuned model, ultimately lowering performance on the fine-tuning dataset. On the other hand, a large temperature skews the interpolation factors toward 0 or 1, which degrades performance on the target dataset. Such extreme values hinder the integration of multiple models, a key requirement for effective generalization to target data distributions. Further details are provided in Section 4.2.

The model achieves the best balance across datasets when $T = 0.01$. This configuration produces the highest total score of 142.8, demonstrating its effectiveness for robust generalization.

## E  QUALITATIVE RESULTS

This section provides an analysis of the qualitative results from the original fc-clip, the standard fine-tuning technique, and the proposed method. Figure D2 shows the output of each method. When evaluated on the source dataset, the standard fine-tuning technique fails to recognize the *backpack*, losing information from the source dataset. On the fine-tuning dataset, the original fc-clip fails to identify key elements such as *road* and *person*. This highlights that OVS models perform well only within the data distribution of the source dataset. When evaluated on the target dataset, the standard fine-tuning technique fails to recognize *ceiling*, a class that does not exist in both the source dataset and the fine-tuning dataset. In contrast, the proposed method successfully identifies both previous and newly learned classes, as well as classes not present in either training dataset.

## F  ADDITIONAL LIMITATIONS

Our method generates unique model weights for each input sample, which makes it challenging to use when the batch size exceeds one. This limitation is also observed in other continual learning approaches (Wang et al., 2022a; Smith et al., 2023; Jin et al., 2023). One potential solution is to apply parallel processing only during the encoder stage. The encoder stage of OVS models generally requires significant computational resources. However, since our method focuses on decoder weight interpolation, multiple samples can be processed in parallel during the encoder stage. Afterward, the embeddings from each sample can be passed through decoders with different weights. While this approach resolves the batch size limitation, the increased computational cost in the decoder stage remains a concern compared to traditional OVS models. To address this issue, further research is needed to develop a parallel processing mechanism for our method.

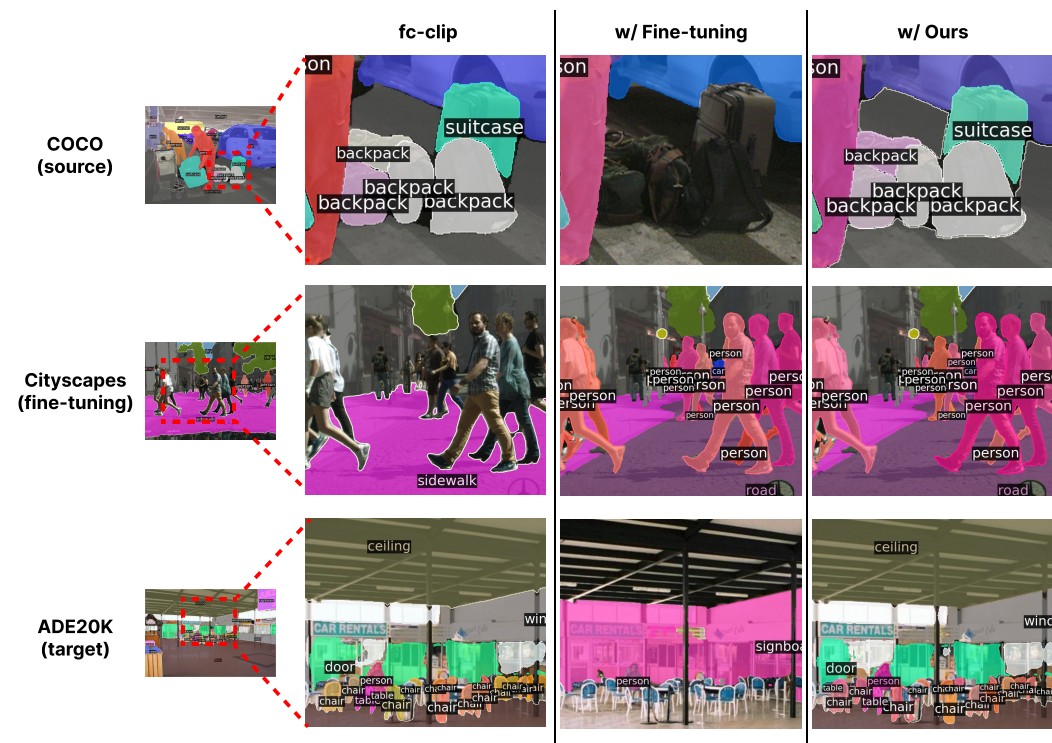

Figure D2: We provide a qualitative analysis on COCO, Cityscapes, and ADE20K. The comparison involves three methods: fc-clip, fine-tuning, and our approach. Both fine-tuning and our method use the Cityscapes dataset to fine-tune fc-clip.

---

**Algorithm 1** Inference Process of Our Method.

---

**Input:** Input $(x_{img}, x_{text})$, encoder $f_{img}$ & $f_{text}$, decoder $f_{dec}$, pre-trained decoder weight $\theta_{dec,pr}$, fine-tuned decoders weight $\{\theta_{dec,ft}^1, \theta_{dec,ft}^2, ..., \theta_{dec,ft}^{N_{ft}}\}$, mean and covariance matrix $\{(\mu_{img}^i, \Sigma_{img}^i, \mu_{text}^i, \Sigma_{text}^i)\}_{i=0}^{N_{ft}}$, pdf of the MVN distribution $p$.

**Output:** Mask & class pairs $\{(m_i, c_i)\}_{i=1}^{N_q}$

**Step 1.** Extract embedding vectors $z$.

$z_{img} \leftarrow f_{img}(x_{img})$
$z_{text} \leftarrow f_{text}(x_{text})$

**Step 2.** Calculate the likelihood l for all data distributions.

$\mathbf{l}_{img} \leftarrow \{p(z_{img}|\mu_{img}^i, \Sigma_{img}^i)\}_{i=0}^{N_{ft}}$
$\mathbf{l}_{text} \leftarrow \{p(z_{text}|\mu_{text}^i, \Sigma_{text}^i)\}_{i=0}^{N_{ft}}$

**Step 3.** Apply softmax and maximum to get $\boldsymbol{\lambda}$.

$\mathbf{s}_{img} \leftarrow \text{softmax}(\mathbf{l}_{img})$
$\mathbf{s}_{text} \leftarrow \text{softmax}(\mathbf{l}_{text})$
$\boldsymbol{\lambda} \leftarrow \text{maximum}(\mathbf{s}_{img}, \mathbf{s}_{text})$

**Step 4.** Interpolate the decoders weight.

$\theta_{dec,new} \leftarrow \theta_{dec,pr} + \sum_{i=1}^{N_{ft}} \lambda_i * (\theta_{dec,ft}^i - \theta_{dec,pr})$

**Step 5.** Compute the output from the decoder.

$(m_i, c_i)_{i=1}^{N_q} \leftarrow f_{dec}(z_{img}, z_{text}; \theta_{dec,new})$

**return** $\{(m_i, c_i)\}_{i=1}^{N_q}$

---

Table A3: Performance comparison between original fine-tuning, previous continual learning methods, and our method. All methods fine-tune the models using Cityscapes. We use PQ, mAP, mIoU for evaluation metrics.

| Method | Fine-tuning Dataset | COCO (previous training) | | | | Cityscapes (fine-tuning) | | | | ADE20K (unseen) | | | |
|---|---|---|---|---|---|---|---|---|---|---|---|---|---|
| | | PQ | mAP | mIoU | Avg | PQ | mAP | mIoU | Avg | PQ | mAP | mIoU | Avg |
| fc-clip | - | 50.1 | 41.1 | 52.0 | 47.7 | 44.0 | 26.8 | 56.2 | 42.4 | 23.5 | 17.1 | 30.4 | 23.7 |
| + Fine-tuning | | -22.7 | -16.2 | -11.8 | -16.9 | +20.1 | +13.9 | +21.2 | +18.4 | -10.3 | -6.3 | -3.9 | -6.8 |
| + ER | | -1.6 | -2.7 | +0.2 | -1.4 | +19.0 | +13.0 | +20.1 | +17.4 | +0.3 | -3.5 | +0.9 | -0.8 |
| + LwF | Cityscapes | -10.7 | -11.9 | -7.9 | -10.2 | +12.2 | +2.7 | +10.2 | +8.3 | -0.8 | -5.4 | +0.8 | -1.8 |
| + EWC | | -25.9 | -19.0 | -13.3 | -19.4 | +19.3 | +11.2 | +18.4 | +16.3 | -9.8 | -8.4 | -4.2 | -7.5 |
| + ECLIPSE | | -6.0 | -6.2 | -3.9 | -5.3 | +2.2 | +0.2 | +4.3 | +2.2 | +0.9 | -3.6 | +2.0 | -0.3 |
| **+ Ours** | | **+0.3** | **+0.5** | **+0.1** | **+0.3** | **+20.2** | **+13.9** | **+21.3** | **+18.5** | **+2.5** | **-1.2** | **+2.5** | **+1.3** |
| X-Decoder | - | 56.7 | 46.9 | 67.4 | 57.0 | 36.3 | 25.4 | 52.9 | 38.2 | 16.7 | 11.7 | 24.9 | 17.8 |
| + Fine-tuning | Cityscapes | -50.4 | -32.2 | -53.7 | -45.5 | +26.6 | +11.7 | +26.7 | +21.7 | -12.9 | -8.1 | -19.7 | -13.5 |
| **+ Ours** | | **-0.4** | **-0.4** | **-0.3** | **-0.3** | **+26.6** | **+11.6** | **+26.7** | **+21.7** | **+0.1** | **+0.5** | **-0.3** | **+0.1** |

Table A4: Performance comparison between original fine-tuning, previous continual learning methods, and our method. All methods fine-tune the models using ADE20K. We use PQ, mAP, mIoU for evaluation metrics.

| Method | Fine-tuning Dataset | COCO (previous training) | | | | ADE20k (fine-tuning) | | | | Cityscapes (unseen) | | | |
|---|---|---|---|---|---|---|---|---|---|---|---|---|---|
| | | PQ | mAP | mIoU | Avg | PQ | mAP | mIoU | Avg | PQ | mAP | mIoU | Avg |
| fc-clip | - | 50.1 | 41.1 | 52.0 | 47.7 | 23.5 | 17.1 | 30.4 | 23.7 | 44.0 | 26.8 | 56.2 | 42.4 |
| + Fine-tuning | | -7.7 | -6.2 | -2.7 | -5.5 | +24.1 | +19.0 | +22.0 | +21.7 | -3.0 | -2.8 | +2.9 | -1.0 |
| + ER | | +0.4 | -0.3 | +2.9 | +1.0 | +21.5 | +16.3 | +19.5 | +19.1 | -3.5 | -2.8 | -1.0 | -2.4 |
| + LwF | ADE20K | -3.8 | -7.1 | -2.4 | -4.4 | +13.7 | +8.4 | +11.3 | +11.1 | -1.0 | -6.2 | -3.0 | -3.4 |
| + EWC | | -11.1 | -9.3 | -6.0 | -8.8 | +20.7 | +16.2 | +18.0 | +18.3 | -2.6 | -3.2 | +0.3 | -1.8 |
| + ECLIPSE | | -0.5 | -1.2 | +0.6 | -0.3 | +0.2 | -0.3 | +3.0 | +1.0 | -5.9 | -4.0 | -2.2 | -4.0 |
| **+ Ours** | | **+1.7** | **+1.4** | **+3.2** | **+2.1** | **+23.8** | **+18.6** | **+21.1** | **+21.2** | **-0.3** | **-0.7** | **+0.6** | **-0.1** |
| X-Decoder | - | 56.7 | 46.9 | 67.4 | 57.0 | 16.7 | 11.7 | 24.9 | 17.8 | 36.3 | 25.4 | 52.9 | 38.2 |
| + Fine-tuning | ADE20K | -37.3 | -33.6 | -42.4 | -37.8 | +28.2 | +18.6 | +27.2 | +24.6 | -3.7 | -9.4 | -0.8 | -4.6 |
| **+ Ours** | | **-1.5** | **-1.7** | **-1.1** | **-1.4** | **+29.2** | **+19.0** | **+27.5** | **+25.2** | **+1.4** | **-6.4** | **+3.5** | **-0.5** |

Table A5: Performance comparison between standard fine-tuning and our method. The underlined values indicate the best score for each dataset. We use PQ, mAP, mIoU for evaluation metrics.

| Method | The order of fine-tuning | COCO (previous) | | | ADE20k (fine-tuning 1) | | | Cityscapes (fine-tuning 2) | | |
|---|---|---|---|---|---|---|---|---|---|---|
| | | PQ | mAP | mIoU | PQ | mAP | mIoU | PQ | mAP | mIoU |
| fc-clip | - | 50.1 | 41.1 | 52.0 | 23.5 | 17.1 | 30.4 | 44.0 | 26.8 | 56.2 |
| + Fine-tuning | ADE20k → Cityscapes | 20.8 | 19.5 | 40.0 | 15.4 | 14.2 | 34.9 | 65.2 | 42.3 | 77.6 |
| + Fine-tuning | Cityscapes → ADE20k | 39.3 | 32.4 | 48.3 | 48.3 | 36.3 | 52.1 | 46.0 | 26.4 | 61.5 |
| + Ours | Cityscapes, ADE20k | **51.6** | **42.5** | **55.3** | **47.0** | **35.9** | **51.4** | **64.3** | **40.7** | **77.6** |

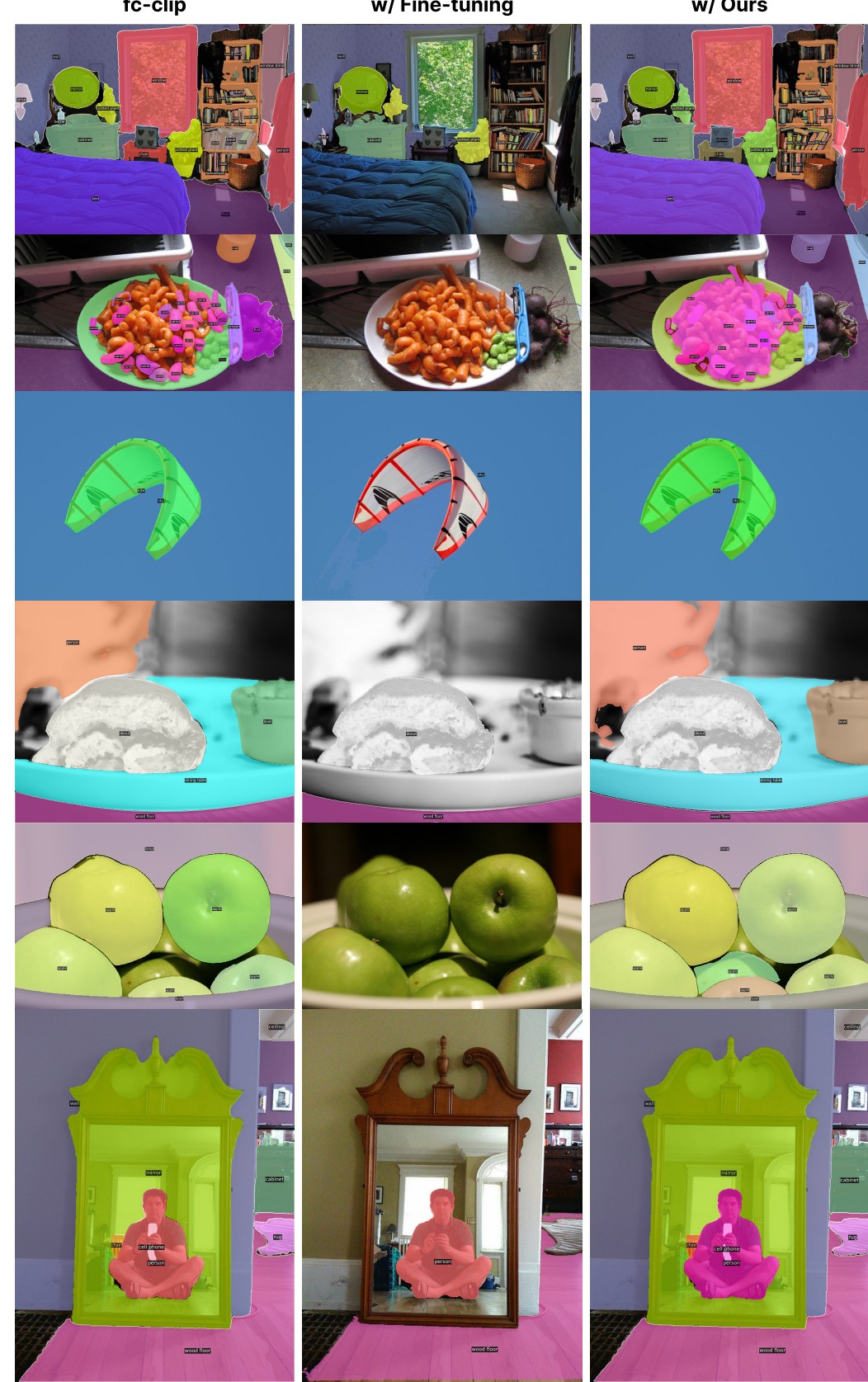

Figure F3: We provide additional qualitative analysis on COCO (previous training dataset). The comparison involves three methods: fc-clip, fine-tuning, and our approach. Fine-tuning and our method both use the Cityscapes dataset to fine-tune fc-clip.

**fc-clip**  **w/ Fine-tuning**  **w/ Ours**

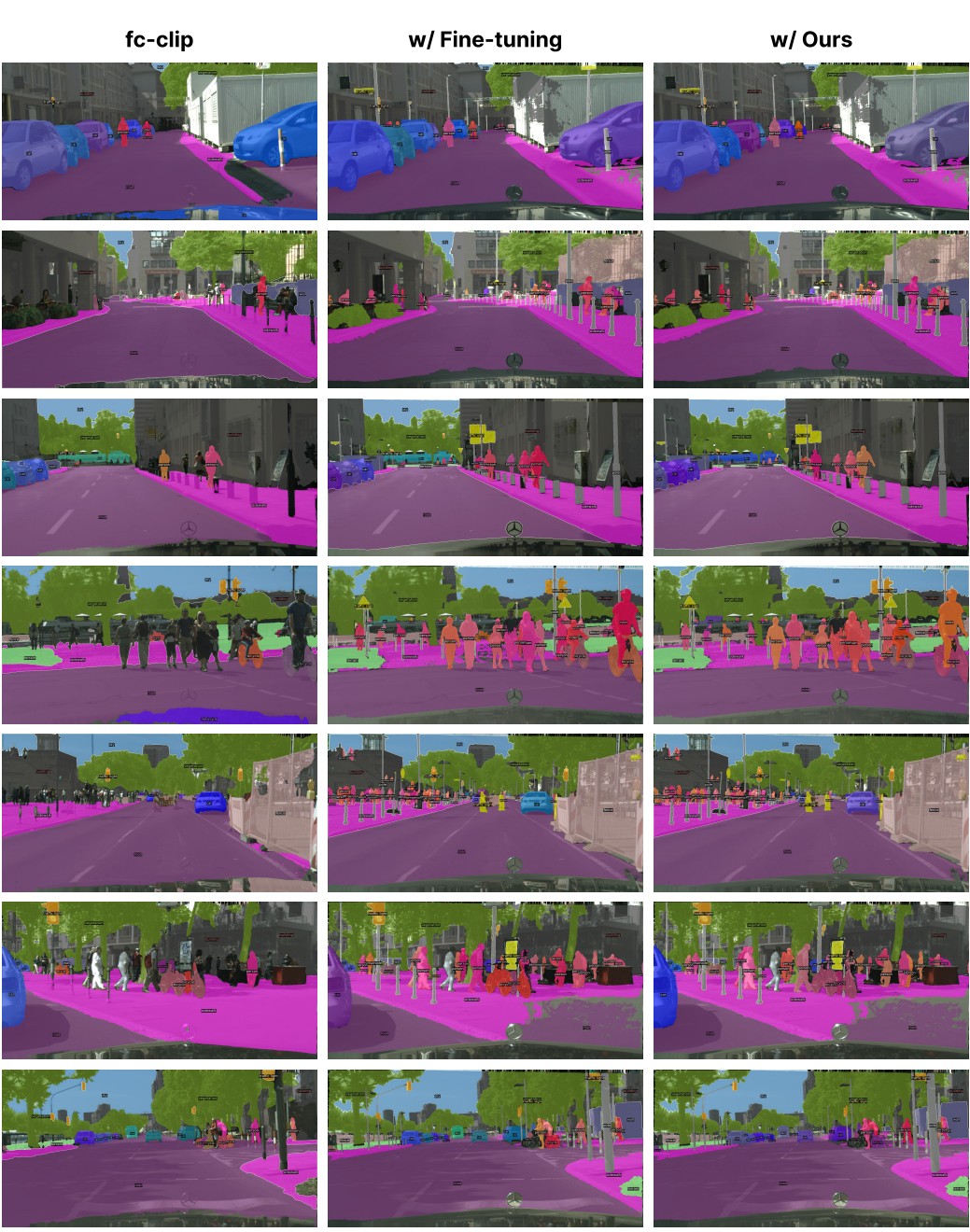

Figure F4: We provide additional qualitative analysis on Cityscapes (fine-tuning dataset). The comparison involves three methods: fc-clip, fine-tuning, and our approach. Fine-tuning and our method both use the Cityscapes dataset to fine-tune fc-clip.

**fc-clip**                **w/ Fine-tuning**                **w/ Ours**

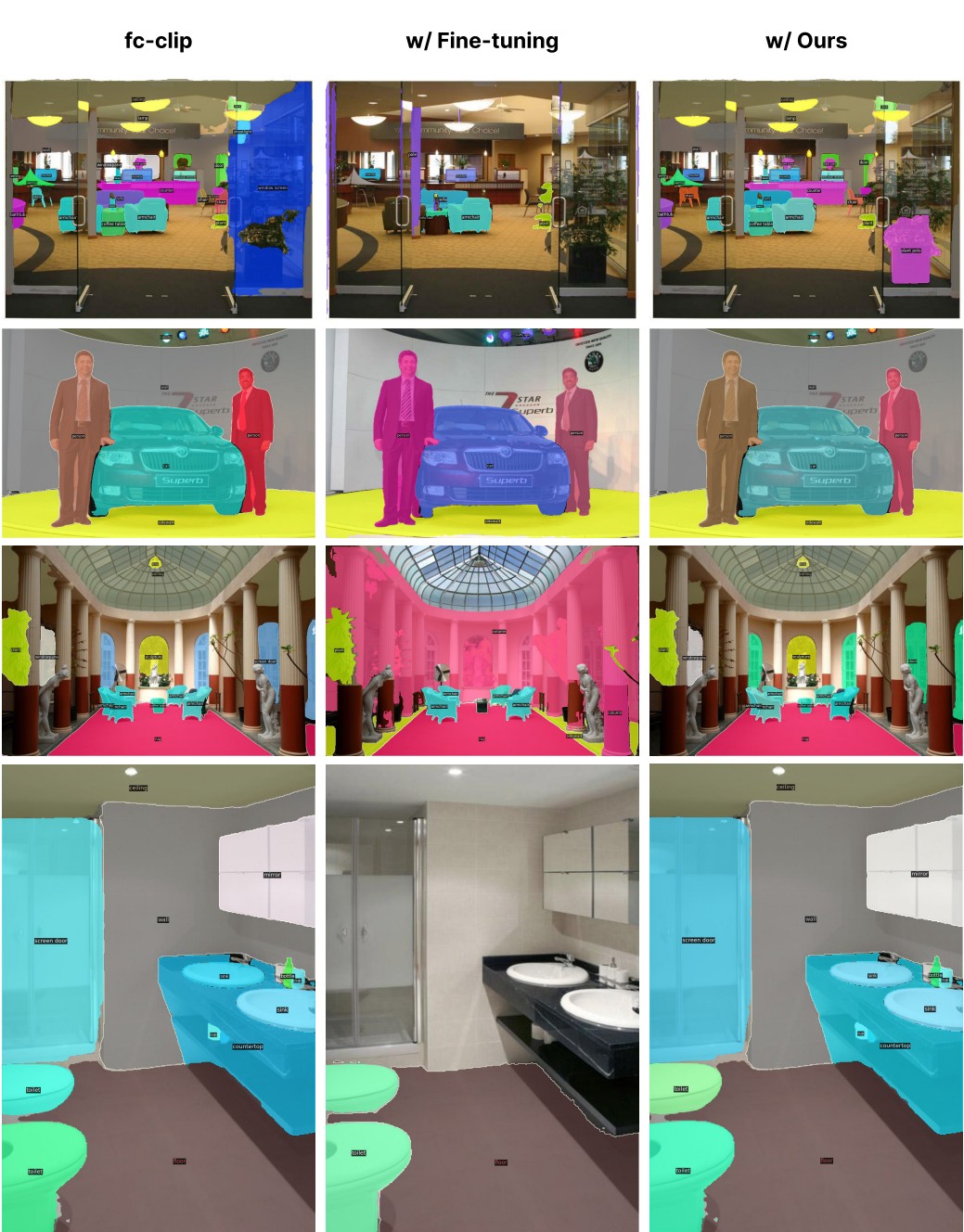

Figure F5: We provide additional qualitative analysis on ADE20K (unseen dataset). The comparison involves three methods: fc-clip, fine-tuning, and our approach. Fine-tuning and our method both use the Cityscapes dataset to fine-tune fc-clip.

