# OpenReview forum: "Overcoming Domain Limitations in Open-vocabulary Segmentation"
_ICLR.cc/2025/Conference — Submitted to ICLR 2025_

### Official Review · Reviewer_xh2L · 2024-10-26

**Soundness:** 3
**Presentation:** 3
**Contribution:** 2
**Rating:** 6
**Confidence:** 3

**Summary:**

This paper addresses the challenge of open-vocabulary segmentation (OVS) models experiencing significant performance drops when applied to unseen domains. The authors propose a method to enable OVS models to learn from new domains while retaining previously acquired knowledge, thereby mitigating catastrophic forgetting.

**Strengths:**

1. The paper effectively addresses the limitation of open-vocabulary segmentation (OVS) models: their poor performance on unseen domains.

2. The method's effectiveness is validated across a diverse range of datasets, showing its robustness and generalizability.

**Weaknesses:**

1. While the proposed method is tested on several datasets, the diversity of domains used in the experiments might still be limited. More extensive testing on a wider range of domains, including those with significant differences from the training data, e.g. GTA5, Night Scenes. This would provide a more comprehensive evaluation of the method's robustness.

2. The use of multivariate normal distributions (MVNs) to measure domain proximity adds complexity to the inference process. The computational overhead and potential latency introduced by this step should be thoroughly analyzed and discussed, especially for real-time applications.

3. The paper does not provide a detailed analysis of how sensitive the proposed method is to hyperparameters, such as the interpolation factors and the parameters of the MVNs.

**Questions:**

Please refer to Weakness

---

> ### Author Response · Authors · 2024-11-26
> **To Reviewer xh2L (1/4)**
>
> Thank you for your positive feedback and for recognizing the strengths of our paper. We deeply appreciate your acknowledgment of how effectively our work addresses the limitations of open-vocabulary segmentation (OVS) models, particularly their poor performance on unseen domains. Additionally, your recognition of the method's robustness and generalizability across diverse datasets aligns closely with our primary goals in this research. Your thoughtful remarks motivate us to continue refining and expanding our approach. Below, we address your concerns in detail.

---

> ### Author Response · Authors · 2024-11-26
> **To Reviewer xh2L (2/4)**
>
> ### **Q1) While the proposed method is tested on several datasets, the diversity of domains used in the experiments might still be limited. More extensive testing on a wider range of domains, including those with significant differences from the training data, e.g. GTA5, Night Scenes. This would provide a more comprehensive evaluation of the method's robustness.**
>
> We agree with the reviewer's suggestion. Evaluating our method using unseen datasets that differ significantly from the domain of the training dataset can comprehensively demonstrate the generalizability of our proposed approach. To this end, we conducted experiments comparing the performance of our method and the baseline on the following datasets: DarkZurich and FoggyZurich, which consist of driving scenes under adverse conditions of nighttime and fog, and GTA5, a synthetic driving scene dataset. The experiments were conducted under the following settings:
>
> - **Backbone**: fc-clip
> - **Previous Training Dataset**: ADE20K
> - **Fine-Tuning Dataset**: COCO
> - **Evaluation Dataset**: GTA5, DarkZurich, FoggyZurich
>
> As shown in the table below, fine-tuning the fc-clip model results in a significant performance drop across all three datasets. This outcome indicates that fine-tuning OVS models degrades performance under adverse conditions or in synthetic domains. In contrast, the proposed method significantly improves the performance of the fc-clip model on these datasets. The fact that the performance surpasses both the fine-tuned model and the original model demonstrates that our adaptive approach to combining the two models is highly effective for unseen samples with substantial domain differences.
>
> | Method          | GTA5   | DarkZurich | FoggyZurich |
> |-----------------|---------|------------|-------------|
> | fc-clip         | 65.6    | 40.2       | 54.4        |
> | + Fine-tuning   | 58.4    | 39.8       | 52.1        |
> | + Ours          | **66.6** | **43.1**   | **55.9**    |
>
> This experiment is important as it demonstrates the effectiveness of our method even when the domain differs substantially from the training dataset, such as in adverse conditions like nighttime or fog, or in synthetic conditions like GTA5. We acknowledge that this experiment provides a more rigorous validation of the proposed method's effectiveness. Accordingly, we have added relevant discussions and experimental results in Section 5.2. Thank you for your constructive questions.

---

> ### Author Response · Authors · 2024-11-26
> **To Reviewer xh2L (3/4)**
>
> ### **Q2) The use of multivariate normal distributions (MVNs) to measure domain proximity adds complexity to the inference process. The computational overhead and potential latency introduced by this step should be thoroughly analyzed and discussed, especially for real-time applications.**
>
> We analyze the computational overhead and latency of each component in our method. We demonstrate that while our method requires additional computational resources, the increase in overall inference time is minimal; unlike other continual learning techniques, our approach does not require additional parameters as the number of learned datasets grows [1], nor does it involve multiple forward passes [2, 3]. Moreover, by applying weight interpolation exclusively to the decoder in encoder-decoder models, our method ensures computational efficiency.
>
> ---
>
> **Inference Time Analysis**
>
> | **Number of Seen Datasets** | **Encoder (ms)** | **Interpolation Factor Estimator (ms)** | **Decoder Weight Interpolation (ms)** | **Decoder (ms)** | **Total Inference Time Per Sample (ms)** | **Change (%)** |
> |-----------------------------|------------------|-----------------------------------------|---------------------------------------|------------------|-------------------------------------------|----------------|
> | 1                           | 97.69           | -                                       | -                                     | 102.30           | 199.99                                   | +0.00%       |
> | 2                           | 97.69           | 0.81                                    | 10.69                                 | 102.30           | 211.48                                   | +5.75%       |
> | 3                           | 97.69           | 1.01                                    | 13.23                                 | 102.30          | 214.23                                   | +7.12%       |
>
> Based on these results, training with two datasets led to a 5.75 percentage point increase in inference time compared to the baseline, while adding an additional dataset resulted in only a 1.37 percentage point increase. Specifically, the interpolation factor estimator, which measures domain proximity, has minimal impact on the overall inference time (+0.4%p when training with two datasets). This indicates that the additional computational cost introduced by our method is negligible compared to the inference time of the encoder and decoder.
>
> **Comparison with PEFT [1]**
>
> | **# Seen Datasets** | **PEFT [1] (ms)**      | **Ours (ms)**        |
> |---------------------|------------------------|----------------------|
> | 1                   | 199.99 (+0.00%)       | 199.99 (+0.00%)     |
> | 2                   | +36.04 (+18.02%)      | +11.49 (+5.75%)     |
> | 3                   | +84.73 (+42.37%)      | +14.24 (+7.12%)     |
>
> We also compared the inference time of our method with an existing continual learning technique [1] designed for closed-set segmentation. Using the COCO validation set, we measured the average inference time per sample, as detailed in the table above. Our method demonstrated faster inference times than PEFT [1] and showed smaller increases in inference time as the number of seen datasets grew.
>
> **Storage Demands**
>
> Our method requires saving the parameters of the learned decoders for each dataset. However, since only the decoder parameters are stored (rather than the entire model), the storage demand is significantly lower compared to other model ensemble approaches [2, 3].
>
> Specifically, our method stores only *6.11% of the total model size* (approximately 80MB), which greatly reduces the storage requirements for model merging. This ensures that our approach does not impose a substantial storage overhead relative to the size of the full model. We have included this discussion in the revised manuscript.
>
> ---
>
> We recognized that the computational overhead introduced by each component of the proposed method was not analyzed in the previous manuscript. Accordingly, we have incorporated the results of the above experiments and analysis in Section 5.3. Thank you for the constructive feedback.
>
>
> #### **References**
>
> 1. Kim, Beomyoung, Joonsang Yu, and Sung Ju Hwang. *"ECLIPSE: Efficient Continual Learning in Panoptic Segmentation with Visual Prompt Tuning."* CVPR, 2024.
> 2. Wortsman, Mitchell, et al. *"Model soups: averaging weights of multiple fine-tuned models improves accuracy without increasing inference time."* International Conference on Machine Learning (ICML), PMLR, 2022.
> 3. Khirodkar, Rawal, et al. *"Sequential Ensembling for Semantic Segmentation."* arXiv preprint arXiv:2210.05387, 2022.

---

> ### Author Response · Authors · 2024-11-26
> **To Reviewer xh2L (4/4)**
>
> ### **Q3) The paper does not provide a detailed analysis of how sensitive the proposed method is to hyperparameters, such as the interpolation factors and the parameters of the MVNs.**
>
> We acknowledge the absence of a sensitivity analysis for the hyperparameter used in our proposed method and have conducted additional experiments to address this issue. While our method does not involve additional hyperparameters related to the MVN distributions, there is a temperature hyperparameter within the softmax operation when calculating the interpolation factor. To evaluate its effect, we performed an ablation study, and the results are presented in the table below.
>
> | **Temperature (T)** | **Previous (COCO)** | **Fine-tuning (ADE20K)** | **Unseen (Cityscapes)** | **Total** |
> |----------------------|---------------------|--------------------------|-------------------------|-----------|
> | 0.0001               | 50.7               | 35.4                    | 43.8                   | 129.9     |
> | 0.001                | 51.2               | 42.2                    | **43.9**                   | 137.3     |
> | 0.01                 | **51.8**           | 47.3                | 43.7                   | **142.8** |
> | 0.1                  | 51.3               | **47.5**                | 43.2                   | 142.0     |
> | 1.0                  | 51.2               | 47.4                    | 43.2                   | 141.8     |
>
>
> #### **Key Observations**
> **When the softmax temperature is small**:
> Performance on the fine-tuning dataset decreases significantly. For optimal performance on the fine-tuning dataset, the interpolation factor multiplied by the weights of the fine-tuned model must be close to 1. However, lower temperature values excessively smooth the interpolation factor, preventing the fine-tuned model from being fully utilized. This phenomenon is also observed in the performance on the previous training dataset.
>
> **When the softmax temperature is large**:
> Performance on the unseen dataset slightly decreases. When the domain of the input sample is uncertain, appropriately interpolating the weights of multiple models is crucial to ensure robust performance across diverse domains. However, higher temperature values cause the interpolation factor to skew toward 0 or 1, leading to reduced performance on the unseen dataset. For a more detailed explanation, please refer to Section 4.2 Discussion.
>
> ---
>
> When the softmax temperature is set to 0.01, the model achieves the most balanced performance across all three datasets, resulting in the highest total score. These findings, along with our analysis, have been incorporated in Appendix D. Thank you for your valuable feedback.

---

> > ### Comment · Reviewer_xh2L · 2024-11-26
> >
> > Thanks for your detailed responses. I maintain the initial rating.

---

> ### Author Response · Authors · 2024-11-26
> **Grateful for Your Constructive Feedback**
>
> Dear Reviewer xh2L,
>
> We deeply appreciate your time and effort in providing feedback on our paper. Your insights have been instrumental in helping us refine and improve our work.
>
> If there are any additional concerns or aspects that require further attention, we would be glad to address them promptly.
>
> Thank you once again for your guidance.
>
>
> Best regards,
>
> The Authors

---

### Official Review · Reviewer_pQF1 · 2024-10-30

**Soundness:** 2
**Presentation:** 3
**Contribution:** 2
**Rating:** 5
**Confidence:** 4

**Summary:**

The paper addresses the issue of forgetting during the fine-tuning process for models on the open vocabulary task by proposing a weight interpolation method based on domain similarity, dynamically determining model parameters for each sample during testing.

**Strengths:**

1. The paper presents a clear and easily understandable approach.
2. Experiments have been conducted on multiple datasets and tasks.

**Weaknesses:**

1. The method proposed in the paper is designed for the open vocabulary task, but some of the experiments in the introduction and ablation sections are based on results on fine-tuning close-vocabulary datasets which lacks persuasiveness.

2. The most important evaluation criterion for the open vocabulary task is performance on unseen data, rather than seen data. The model's performance on unseen ADE and Cityscapes datasets shows no significant advantage over the original FC-CLIP, and in some metrics, it even declines.

3. The method is simply an ensemble of weights, which is too straightforward and lacks significant insight.

4. The requirement to save model parameters for each dataset places a high demand on storage.

5. The unseen dataset contains both unseen and seen categories. The performance improvement of the method on unseen data compared to FC-CLIP is likely due to the seen categories, as the capability for unseen categories largely depends on the original CLIP. This is evident from Figure 8, where the original FC-CLIP is already able to segment the unseen category "ceiling." Therefore, the method does not seem to have a significant impact on the open vocabulary task.

**Questions:**

1. It is hard to say the method is specifically designed for the open-vocabulary task. In other words, can the proposed method also be applied to traditional continual learning tasks?

---

> ### Author Response · Authors · 2024-11-25
> **Response to Reviewer pQF1 (1/8)**
>
> Thank you for highlighting the strengths of our paper, particularly the clarity and comprehensibility of our approach, as well as the comprehensive evaluation across multiple datasets and tasks. These aspects were central to our efforts in making the method both accessible and widely applicable. Below, we address your concerns in detail.

---

> ### Author Response · Authors · 2024-11-25
> **Response to Reviewer pQF1 (2/8)**
>
> ### **Q1) The method is simply an ensemble of weights, which is too straightforward and lacks significant insight.**
>
> The proposed method of ensembling model weights for open-vocabulary segmentation (OVS) models may seem straightforward. However, we argue that this simplicity does not diminish the insights provided by our paper.
>
> Notably, the scenario we address is distinct from those explored in prior research. Specifically, we identify a critical need for methods that enhance existing OVS models to perform effectively in additional, user-specified domains while preserving their open-vocabulary capabilities. This requirement is particularly relevant in resource-constrained settings, yet no existing work has addressed this challenge. Our paper aims to fill this gap by offering a practical solution.
>
> ---
>
> ### **Key Contributions**
>
> Our paper makes the following contributions:
>
> 1. **Identifying the limitation** of OVS models working effectively only within their training dataset domains.
> 2. **Demonstrating the impact of fine-tuning**, which significantly degrades open-vocabulary performance due to catastrophic forgetting.
> 3. **Thoroughly evaluating the limitations** of applying existing continual learning methods to this problem in OVS models.
> 4. **Proposing a novel method** that resolves these issues.
> 5. **Introducing a new evaluation protocol** tailored to this scenario.
>
> These contributions offer a fresh perspective on the limitations of OVS models and propose a novel solution, ensuring our paper provides substantial academic insights and establishes a new research direction in the OVS field. Detailed explanations for each point are provided below.
>
> ---
>
> #### **1) Identifying the Domain-Specific Limitations of OVS Models**
>
> We demonstrated that OVS models are effective only within the domains of their training datasets. *Table 1* highlights the significant performance gap between state-of-the-art (SOTA) OVS models and closed-set segmentation models when evaluated on domains different from their training datasets.
>
> Such a disparity suggests that OVS models are not practical for real-world applications, where recognizing new object classes in unseen domains is critical. Addressing this limitation requires training techniques that adapt OVS models to perform well in user-specified domains—an issue unexplored in previous OVS research. Identifying and emphasizing this limitation alone provides valuable insights for readers.
>
> ---
>
> #### **2) Highlighting the Challenges of Fine-Tuning Due to Catastrophic Forgetting**
>
> To address the first issue, we applied *fine-tuning*—a straightforward and intuitive solution. However, we discovered that fine-tuning OVS models on new datasets leads to *catastrophic forgetting*, significantly degrading their performance on previously learned tasks.
>
> While this result may seem predictable, no prior research has systematically evaluated the negative effects of fine-tuning on OVS models. Through rigorous analysis, we demonstrated that this problem is not easily solvable with simple methods, highlighting the need for dedicated research to tackle this challenge.
>
> ---
>
> #### **3) Demonstrating the Limitations of Applying Existing Continual Learning Methods**
>
> To address catastrophic forgetting, we applied existing *continual learning methods* to OVS models. While these methods are commonly proposed solutions for such problems, our thorough experiments revealed their limitations in this context.
>
> Specifically:
> - We analyzed how techniques like *parameter-efficient fine-tuning (PEFT)* and other continual learning approaches fail when applied to OVS models.
> - We also evaluated *joint training*, which combines all datasets at once, and showed that it does not resolve the issues in our assumed scenario.
>
> By presenting these results, we emphasize the necessity of novel approaches to address the limitations of existing methods in OVS contexts.
>
>
>
> *The following content is continued below.*

---

> ### Author Response · Authors · 2024-11-25
> **Response to Reviewer pQF1 (3/8)**
>
> #### **4) Proposing a Novel Method to Address the Problem**
>
> We proposed a new method based on *per-sample domain estimation and weight interpolation* to address the identified challenges. While this approach may seem trivial at first glance, it distinguishes itself from prior methods.
>
> - **Dynamic Weight Interpolation**: Unlike fixed-weight ensembling approaches [1, 2], our method dynamically interpolates between fine-tuned and pre-trained decoders based on the input sample.
> - **Efficient Memory Usage**: We focused on interpolating weights specifically within the decoder of segmentation models, requiring the storage of only *6.11% of the model's total capacity*. This technique offers a novel approach to segmentation model ensembling, setting it apart from previous studies [3, 4].
> - **Fast Inference**: This design ensures *faster inference times* compared to PEFT approaches.
>
> ---
>
> #### **5) Introducing a New Evaluation Protocol for the Scenario**
>
> The scenario we propose has not been addressed in prior research, necessitating the creation of a new evaluation protocol. To this end, we designed a novel evaluation protocol by integrating methodologies from existing continual learning and OVS model literature. While we acknowledge that the submission version had several shortcomings, we were able to address many of them during the Rebuttal period based on the reviewers' feedback.
>
> For instance, in the original submission, we used a fixed order of the three training datasets—COCO, Cityscapes, and ADE20K. Now, we consider all possible sequential training orders for these datasets. Additionally, in the testing phase, we evaluate performance across (1) previously seen training datasets, (2) fine-tuning scenarios, and (3) unseen datasets. Notably, we have expanded the unseen datasets to include adverse condition driving scene datasets such as DarkZurich and FoggyZurich, as well as synthetic datasets like GTA5, to enable a more detailed evaluation of OVS models. Thanks to the reviewers' suggestions, we have developed a much more comprehensive evaluation protocol, for which we are sincerely grateful. We also believe this evaluation protocol represents a significant contribution of our paper.
>
> ---
>
> #### **Conclusion**
>
> We hope this response addresses the reviewers’ concerns. We acknowledge that our initial explanation may not have been sufficient and have incorporated the above discussion in Section 1 to improve clarity. Please feel free to reach out if there are any further questions.
>
> ---
>
> #### **References**
>
> 1. Ilharco, Gabriel, et al. *"Editing models with task arithmetic."* ICLR, 2023.
> 2. Wortsman, Mitchell, et al. *"Model soups: averaging weights of multiple fine-tuned models improves accuracy without increasing inference time."* ICML, 2022.
> 3. Khirodkar, Rawal, et al. *"Sequential Ensembling for Semantic Segmentation."* arXiv preprint arXiv:2210.05387, 2022.
> 4. Dang, Truong, et al. *"Two-layer Ensemble of Deep Learning Models for Medical Image Segmentation."* Cognitive Computation, 2024.

---

> ### Author Response · Authors · 2024-11-25
> **Response to Reviewer pQF1 (4/8)**
>
> ### **Q2) The most important evaluation criterion for the open vocabulary task is performance on unseen data, rather than seen data. The model's performance on unseen ADE and Cityscapes datasets shows no significant advantage over the original FC-CLIP, and in some metrics, it even declines.**
>
> As discussed earlier, the goal of our method is to improve the performance of OVS models on fine-tuning datasets while preserving performance on unseen datasets. Therefore, the fact that performance gains on unseen datasets are occasionally small does not undermine the validity of our approach. As shown in *Table 2*, performance degradation on unseen datasets is significantly larger for fine-tuning and continual learning methods compared to our approach.
>
> ---
>
> #### **Table 2: Performance comparison between standard fine-tuning, previous continual learning methods, and our method.**
>
> **(a) Cityscapes**
>
> | Method         | COCO      | Cityscapes      | ADE20K     |
> |----------------|-----------|-----------------|------------|
> |                | previous  | fine-tuning     | unseen     |
> | fc-clip        | 50.1      | 44.0            | 23.5       |
> | + Fine-tuning  | -22.7     | +20.1           | -10.3      |
> | + ER           | -1.6      | +19.0           | +0.3       |
> | + LwF          | -10.7     | +12.2           | -0.8       |
> | + EWC          | -25.9     | +19.3           | -9.8       |
> | + ECLIPSE      | -6.0      | +2.2            | +0.9       |
> | + Ours         | **+0.3**  | **+20.2**       | **+2.5**   |
> | X-Decoder      | 56.7      | 36.3            | 16.7       |
> | + Fine-tuning  | -50.4     | +26.6           | -12.9      |
> | + Ours         | **-0.4**  | **+26.6**       | **+0.1**   |
>
>
> **(b) ADE20K**
>
> | Method         | COCO      | ADE20K          | Cityscapes |
> |----------------|-----------|-----------------|------------|
> |                | previous  | fine-tuning     | unseen     |
> | fc-clip        | 50.1      | 23.5            | 44.0       |
> | + Fine-tuning  | -7.7      | +24.1           | -3.0       |
> | + ER           | +0.4      | +21.5           | -3.5       |
> | + LwF          | -3.8      | +13.7           | -1.0       |
> | + EWC          | -11.1     | +20.7           | -2.6       |
> | + ECLIPSE      | -0.5      | +0.2            | -5.9       |
> | + Ours         | **+1.7**  | **+23.8**       | **-0.3**   |
> | X-Decoder      | 56.7      | 16.7            | 36.3       |
> | + Fine-tuning  | -37.3     | +28.2           | -3.7       |
> | + Ours         | **-1.5**  | **+29.2**       | **+1.4**   |
>
>
> ---
>
> That said, we agree with the reviewer's valid observation that performance on unseen datasets is a critical metric for OVS models. Acknowledging this importance, we conducted additional experiments on 8 unseen datasets commonly used in the OVS field, as presented in *Table 5*. Interestingly, our method not only preserved performance on unseen datasets but also enhanced it overall. We believe this improvement arises as a beneficial side effect of leveraging knowledge learned from fine-tuning datasets to better recognize unseen datasets.
>
> ---
>
> #### **Table 5: Performance comparison across seen and unseen datasets**
>
> | **Method**        | **Order of Fine-Tuning**       | **LVIS (mAP)** | **BDD100K (PQ)** | **Mapillary (mIoU)** | **PC-59 (mIoU)** | **PC-459 (mIoU)** | **PAS-20 (mIoU)** | **PAS-21 (mIoU)** | **A-847 (mIoU)** |
> |--------------------|--------------------------------|----------------|------------------|----------------------|------------------|-------------------|-------------------|-------------------|------------------|
> | **fc-clip**       | -                              | 20.5           | 19.0             | 26.0                 | 53.0             | 16.9              | 93.1              | 80.2              | 13.8             |
> | **+ Fine-tuning**  | Cityscapes → ADE20k            | 21.7           | 19.7             | 27.8                 | 52.1             | 17.2              | 92.3              | 76.7              | 16.0             |
> | **+ Fine-tuning**  | ADE20k → Cityscapes            | 10.4           | 21.3             | 24.2                 | 45.9             | 13.5              | 87.4              | 70.7              | 11.5             |
> | **+ Ours**         | Cityscapes, ADE20k             | **23.1**       | **22.6**         | **29.1**             | **54.9**         | **17.9**          | **93.6**          | **80.7**          | **16.3**         |
>
> ---
>
> We recognize that this point was not clearly articulated in the original manuscript. As a result, we have revised the discussion to clarify this aspect. Thank you for raising this insightful question.

---

> > ### Comment · Reviewer_pQF1 · 2024-11-29
> > **Response to feedback 4/8**
> >
> > If the comparison is based on performance on seen datasets, the baseline should involve joint training on those same datasets. This strategy would likely yield similar performance to your method in seen datasets. From this perspective, the authors' approach does not demonstrate a significant advantage in the OVS setting.

---

> > > ### Author Response · Authors · 2024-11-30
> > > **Response to Additional Feedback from Reviewer pQF1 (2/3)**
> > >
> > > **Q4) If the comparison is based on performance on seen datasets, the baseline should involve joint training on those same datasets. This strategy would likely yield similar performance to your method in seen datasets. From this perspective, the authors' approach does not demonstrate a significant advantage in the OVS setting.**
> > >
> > > | **Method**       | **COCO (previous)** | **Cityscapes (fine-tuning)** | **ADE20K (unseen)** |
> > > |-------------------|-------------------|-----------------------------|---------------------|
> > > | fc-CLIP          | 50.1              | 44.0                        | 23.5                |
> > > | + Fine-tuning    | -22.7             | +20.1                       | -10.3               |
> > > | + Joint Training | +0.6              | +17.9                       | +1.7                |
> > > | + Ours           | +0.3              | +20.2                       | +2.5                |
> > >
> > > | **Method**       | **COCO (previous)** | **ADE20K (fine-tuning)** | **Cityscapes (unseen)** |
> > > |-------------------|-------------------|--------------------------|--------------------------|
> > > | fc-CLIP          | 50.1              | 23.5                     | 44.0                     |
> > > | + Fine-tuning    | -7.7              | +24.1                    | -3.0                     |
> > > | + Joint Training | +1.4              | +16.5                    | -1.2                     |
> > > | + Ours           | +1.7              | +23.8                    | -0.3                     |
> > >
> > > As suggested by the reviewer, we have added the results of joint training to Table 2. The results show that for the previous dataset (COCO), performance is maintained or improved, and it achieves similar performance compared to our method. However, on the fine-tuning dataset, we observed that the performance improvement of joint training is relatively smaller compared to our method or fine-tuning alone. Additionally, for the unseen dataset, joint training performs worse than our method.
> > >
> > > | Method           | The order of fine-tuning | COCO (previous) | ADE20K (fine-tuning 1) | Cityscapes (fine-tuning 2) |
> > > |-------------------|--------------------------|---------------|-------------------------|----------------------------|
> > > | fc-clip          | -                        | 50.1          | 23.5                   | 44.0                       |
> > > | + Fine-tuning    | ADE → City               | 20.8          | 15.4                   | 65.2                       |
> > > | + Fine-tuning    | City → ADE               | 39.3          | 48.3                   | 46.0                       |
> > > | + Joint Training | City, ADE                | 48.6          | 35.5                   | 60.5                       |
> > > | + Ours           | City, ADE                | **51.6**      | **47.0**               | **64.3**                   |
> > >
> > >
> > > | Method           | Previous dataset         | The order of fine-tuning    | LVIS  | BDD100K | Mapillary | PC-59 | PC-459 | PAS-20 | PAS-21 | A-847 |
> > > |-------------------|------------------------|-----------------------------|-------|---------|-----------|-------|--------|--------|--------|-------|
> > > | fc-clip          | COCO                  | -                           | 20.5  | 19.0    | 26.0      | 53.0  | 16.9   | 93.1   | 80.2   | 13.8  |
> > > | + Fine-tuning    | COCO                  | Cityscapes → ADE20k         | 21.7  | 19.7    | 27.8      | 52.1  | 17.2   | 92.3   | 76.7   | 16.0  |
> > > | + Fine-tuning    | COCO                  | ADE20k → Cityscapes         | 10.4  | 21.3    | 24.2      | 45.9  | 13.5   | 87.4   | 70.7   | 11.5  |
> > > | + Joint Training | COCO, Cityscapes, ADE20k | -                           | 21.5  | 21.8    | 28.0      | 53.2  | 17.3   | 93.3   | **80.9** | 15.2  |
> > > | + Ours           | COCO                  | Cityscapes, ADE20k          | **23.1** | **22.6** | **29.1** | **54.9** | **17.9** | **93.6** | 80.7 | **16.3** |
> > >
> > > Additionally, we have included the results of joint training on the three datasets (COCO, Cityscapes, ADE20K) in Tables 4 and 5. As shown in Table 4, joint training achieves higher performance on the previous dataset (COCO) compared to sequential fine-tuning but performs worse than our method across all three datasets. Furthermore, as demonstrated by experiments on eight different target datasets, joint training shows better generalizability than sequential fine-tuning but still underperforms compared to our method. These results and analyses have been added to Section 5.2.
> > >
> > > These results indicate that our method effectively integrates the knowledge of models trained on seen datasets, thereby improving performance even on unseen datasets. Therefore, we respectfully disagree with the opinion that our approach does not demonstrate advantages in the OVS setting.

---

> ### Author Response · Authors · 2024-11-25
> **Response to Reviewer pQF1 (5/8)**
>
> ### **Q3) The unseen dataset contains both unseen and seen categories. The performance improvement of the method on unseen data compared to FC-CLIP is likely due to the seen categories, as the capability for unseen categories largely depends on the original CLIP. This is evident from Figure 8, where the original FC-CLIP is already able to segment the unseen category "ceiling." Therefore, the method does not seem to have a significant impact on the open vocabulary task.**
>
> As mentioned earlier, improving performance on unseen datasets is not the primary goal of our approach. Our main objective is to *preserve* the performance of OVS models on unseen and previously trained datasets while extending their domain capabilities. However, as noted in *Table 4*, there are cases where our method achieves performance improvements on unseen datasets. We believe these improvements result from leveraging knowledge learned from fine-tuning datasets.
>
> ---
>
> #### **Analysis Based on Seen and Unseen Categories**
>
> We agree with the reviewer that analyzing the performance of our method from the perspective of seen and unseen categories is valuable. To investigate this, we conducted experiments comparing the proposed method with the original FC-CLIP, separating performance on seen and unseen classes. Specifically:
> - **Previous Training Dataset**: COCO
> - **Fine-Tuning Dataset**: Cityscapes
> - **Evaluation Dataset**: ADE20K
>
> In this setup:
> - **Unseen Classes**: ADE20K classes not present in COCO or Cityscapes (truly unseen).
> - **Seen Classes**: Classes present in COCO or Cityscapes.
>
> The results are summarized in the table below:
>
> | **Method** | **Seen Classes (mIoU)** | **Unseen Classes (mIoU)** |
> |------------|--------------------------|---------------------------|
> | **fc-clip** | 35.0                    | 28.6                      |
> | **Ours**    | **37.9**                | **30.9**                  |
>
> ---
>
> #### **Key Observations**
>
> The results indicate that the proposed method improves performance on both seen and unseen classes. This suggests that merging the domain-specific knowledge of the two OVS model decoders through our weight interpolation technique truly enhances the generalization capability for unseen datasets.
>
> While enhancing performance on unseen classes is not the core objective of our work, we acknowledge that it is an important consideration when working with OVS models. We believe this additional experiment and analysis significantly improve the quality of our paper by providing a deeper understanding of the method's impact. We have included this discussion in Section 5.2.
>
> Thank you for the constructive feedback. If further experimental results or clarifications are needed, please feel free to let us know.

---

> > ### Comment · Reviewer_pQF1 · 2024-11-29
> > **Response to feedback 5/8**
> >
> > The goal of OVS is to leverage seen datasets to improve performance on unseen datasets. The authors state, "Our main objective is to preserve the performance of OVS models on unseen and previously trained datasets while extending their domain capabilities." If a method can maintain strong performance across all seen datasets, it is likely to also perform well on unseen datasets. Thus, the concept of continual learning in the OVS framework does not significantly differ from traditional continual learning in closed-set scenarios. In my view, the authors have proposed a conventional approach to continual learning within a closed-set context, yet they frame it within the OVS narrative, which lacks novelty.

---

> > > ### Author Response · Authors · 2024-11-30
> > > **Response to Additional Feedback from Reviewer pQF1 (3/3)**
> > >
> > > **Q5) The goal of OVS is to leverage seen datasets to improve performance on unseen datasets. The authors state, "Our main objective is to preserve the performance of OVS models on unseen and previously trained datasets while extending their domain capabilities." If a method can maintain strong performance across all seen datasets, it is likely to also perform well on unseen datasets. Thus, the concept of continual learning in the OVS framework does not significantly differ from traditional continual learning in closed-set scenarios. In my view, the authors have proposed a conventional approach to continual learning within a closed-set context, yet they frame it within the OVS narrative, which lacks novelty.**
> > >
> > > As mentioned earlier, our analysis shows that OVS models may perform well on seen datasets but fail to achieve satisfactory performance on unseen datasets. However, we acknowledge that utilizing more datasets for model training could positively impact performance on unseen datasets. Nevertheless, the availability of datasets is limited, and collecting new datasets is a highly challenging task. Thus, such an approach is not practically feasible. Furthermore, even if this approach were adopted, the improved performance on unseen datasets would likely result from training on a large number of datasets and learning an extensive vocabulary, rather than effectively leveraging the knowledge from seen datasets.
> > >
> > > In contrast, our method focuses on effectively utilizing the knowledge gained from diverse seen datasets to improve performance on unseen datasets, aligning closely with the core objectives of OVS. Specifically, our approach progressively expands the range of seen datasets, broadening the scope of the model’s learned knowledge and driving performance improvements on unseen datasets. These outcomes are further validated by additional experiments conducted based on the reviewer’s feedback (**Table 6**), which demonstrate improved performance even on truly unseen classes within unseen datasets, excluding seen classes.
> > >
> > > #### **Table 6: Comparison of performance on seen and truly unseen classes.**
> > >
> > > | **Method** | **Seen Classes (mIoU)** | **Unseen Classes (mIoU)** |
> > > |------------|--------------------------|---------------------------|
> > > | **fc-clip** | 35.0                    | 28.6                      |
> > > | **Ours**    | **37.9**                | **30.9**                  |
> > >
> > > Moreover, research similar to our approach, which utilizes continual learning to maintain or enhance zero-shot performance, has been actively explored in other domains [1,2,3]. These studies aim to extend models with continual learning while preserving or improving performance on unseen (zero-shot) datasets, distinguishing themselves from traditional continual learning approaches that typically operate in closed-set settings.
> > >
> > > In summary, our method effectively leverages diverse seen datasets to expand the open-vocabulary capacity for unseen categories, clearly differentiating itself from conventional continual learning methods.
> > >
> > > #### **Reference**
> > >
> > > [1] Wu, Tongtong, et al. "Continual learning for large language models: A survey." arXiv preprint arXiv:2402.01364 (2024).
> > > [2] Gowda, Shreyank N., Davide Moltisanti, and Laura Sevilla-Lara. "Continual Learning Improves Zero-Shot Action Recognition." arXiv preprint arXiv:2410.10497 (2024).
> > > [3] Zheng, Zangwei, et al. "Preventing zero-shot transfer degradation in continual learning of vision-language models." In ICCV, 2023.

---

> > > > ### Author Response · Authors · 2024-12-01
> > > > **Kind Follow-Up on Discussion Period Feedback**
> > > >
> > > > Dear Reviewer,
> > > >
> > > > With only two days left in the discussion period, we kindly follow up on our earlier reminders regarding your feedback. We have carefully worked to address the questions and concerns you raised through our responses and revisions, and we believe that we have resolved most, if not all, of the issues you highlighted.
> > > >
> > > > If our efforts and revisions meet your expectations, we would greatly value your acknowledgment. However, if you have any additional concerns, we are more than willing to address them promptly before the discussion period concludes.
> > > >
> > > > Thank you for your time and thoughtful consideration.
> > > >
> > > > Sincerely,
> > > > The Authors

---

> ### Author Response · Authors · 2024-11-25
> **Response to Reviewer pQF1 (6/8)**
>
> ### **Q4) The method proposed in the paper is designed for the open vocabulary task, but some of the experiments in the introduction and ablation sections are based on results on fine-tuning close-vocabulary datasets which lack persuasiveness.**
>
> Thank you for the thoughtful comment. As the reviewer noted, we identified that performance on unseen datasets was missing in some experiments in the introduction and ablation studies.
>
> ---
>
> #### **Updates in Ablation Studies**
>
> To address this, we have included the average performance results evaluated across eight different unseen datasets in Tables 8, 9, and 10, as shown below:
>
> ---
>
> ##### **Table 8: Performance of image-only, text-only, and combined distributions**
>
> || **COCO (previous)** | **Cityscapes (fine-tuning)** | **Unseen** |
> |-------------------|---------------------|-----------------------------|----------------------|
> | Image only        | 51.5               | 43.4       | 40.3                |
> | Text only         | **51.9**               | 60.7          | 40.6                |
> | Image + Text      | 51.6               | **64.3**            | **40.9**                |
>
> ---
>
> ##### **Table 9: Performance comparison of prototype models**
>
> || **COCO (previous)** | **Cityscapes (fine-tuning)** | **Unseen** |
> |------------------------------|---------------------|-----------------------------|----------------------|
> | K-means clustering          | 42.4               | 64.1                        | 40.6                |
> | Kernel density estimation    | 48.1               | 57.4                        | 40.6                |
> | MVN distribution             | **50.4**               | **64.3**                        | **40.9**                |
>
> ---
>
> As shown in *Tables 8* and *9*, our design choices not only demonstrate outstanding effectiveness on fine-tuning datasets but also improve performance on unseen and previous datasets. Specifically, as indicated in *Table 8*, combining image and text distributions achieves the highest performance on fine-tuning datasets while ensuring that performance on previous and unseen datasets does not degrade. Moreover, as highlighted in *Table 9*, the use of MVN distributions consistently delivers strong results across all dataset types. These findings highlight the effectiveness of our design choices across previous, fine-tuning, and unseen scenarios.
>
> ---
>
> ##### **Table 10: Weight Interpolation vs. Prompt-Tuning**
>
> | | **COCO (previous)** | **Cityscapes (fine-tuning)** | **Unseen** |
> |---------------------|---------------------|-----------------------------|----------------------|
> | Prompt              | 43.3               | 48.9                        | 39.1                |
> | Weight Interpolation | **50.4**           | **64.3**                    | **40.9**            |
>
> ---
>
> In *Table 10*, we compared our weight interpolation method with prompt-tuning approaches [1, 2]. The results show that weight interpolation outperforms prompt-tuning not only on fine-tuning datasets but also on unseen and previous datasets. This demonstrates that our method is more effective in preserving and extending the capabilities of OVS models.
>
> ---
>
> #### **Clarifications in the Introduction**
>
> In *Table 1* of the introduction, we compare the performance of fine-tuned closed-set segmentation models (e.g., Mask2Former) with that of non-fine-tuned OVS models. While the reviewer expressed concerns regarding the persuasiveness of this comparison, the experiment was deliberately designed to highlight a key motivation for our study.
>
> Specifically, it aimed to emphasize that fine-tuned conventional segmentation models significantly outperform OVS models on the target dataset. Through this comparison, we intended to illustrate a critical limitation of OVS models: *they often fail to meet practical expectations, even when presumed to be suitable for unseen datasets*. This observation serves as the starting point for the scenario explored in our work, underlining the need for practical solutions to address the challenges faced by OVS models.
>
> We acknowledge that the evaluation dataset is seen for the closed-set model and unseen for the OVS model, which raises a concern about unfair comparison. For further details on this issue, we kindly refer the reviewer to our response to Q2 in Reviewer mCxt's comments, where we address this point more thoroughly.
>
> ---
>
> **Conclusion**
>
> We believe these updates improve the clarity and persuasiveness of our experiments, addressing the reviewer's concerns. Thank you for the constructive feedback, and please let us know if additional experiments or explanations are needed.
>
> ---
>
> **References**
>
> 1. Kim, Beomyoung, Joonsang Yu, and Sung Ju Hwang. *"ECLIPSE: Efficient Continual Learning in Panoptic Segmentation with Visual Prompt Tuning."* CVPR, 2024.
> 2. Wang, Yabin, Zhiwu Huang, and Xiaopeng Hong. *"S-prompts learning with pre-trained transformers: An Occam’s razor for domain incremental learning."* NeurIPS, 2022.

---

> ### Author Response · Authors · 2024-11-25
> **Response to Reviewer pQF1 (7/8)**
>
> ### **Q5) It is hard to say the method is specifically designed for the open-vocabulary task. In other words, can the proposed method also be applied to traditional continual learning tasks?**
>
> As mentioned earlier, our research started with the observation that OVS models perform poorly on domains they have not been trained on, and we proposed a method to address this issue. However, we agree with the reviewer that exploring whether our method can be applied to traditional continual learning tasks is a valuable direction to investigate.
>
> ---
>
> #### **Applicability to Traditional Continual Learning Tasks**
>
> The proposed method leverages the OVS model's encoder to estimate the domain of the input sample, then dynamically interpolates between the decoders of multiple OVS models to adjust the model. This approach can potentially be extended to traditional continual learning tasks.
>
> In traditional continual learning, recent research has investigated methods like prompt-tuning [1] and LoRA [2], which train parameters independently for each incremental session. This results in a unique parameter set for every session, independent of the initial model. Our method, which supports the use of independent parameter sets, can be extended to such environments. Specifically, by utilizing the initial model alongside the models generated in each incremental session, our approach estimates the domain of input samples, merges the corresponding parameters, and enables accurate predictions while avoiding catastrophic forgetting.
>
> ---
>
> #### **Future Work**
>
> Reflecting the reviewer's insightful question, we have included a discussion in Appendix D on how the proposed method can be adapted to traditional continual learning tasks. Furthermore, we plan to explore this extension in greater depth as part of future work.
>
> Thank you for bringing up this important and thoughtful point.
>
> ---
>
> #### **References**
>
> 1. Smith, James Seale, et al. *"Coda-prompt: Continual decomposed attention-based prompting for rehearsal-free continual learning."* CVPR, 2023.
> 2. Liang, Yan-Shuo, and Wu-Jun Li. *"InfLoRA: Interference-Free Low-Rank Adaptation for Continual Learning."* CVPR, 2024.

---

> ### Author Response · Authors · 2024-11-25
> **Response to Reviewer pQF1 (8/8)**
>
> ### **Q6) The requirement to save model parameters for each dataset places a high demand on storage.**
>
> Our method requires saving the parameters of the learned decoders for each dataset. However, since only the decoder parameters are stored (rather than the entire model), the storage demand is significantly lower compared to other model ensemble approaches [1, 2].
>
> Specifically, our method stores only *6.11% of the total model size* (approximately 80MB), which greatly reduces the storage requirements for model merging. This ensures that our approach does not impose a substantial storage overhead relative to the size of the full model. We have included this discussion in Section 5.3.
>
> ---
>
> #### **References**
>
> 1. Wortsman, Mitchell, et al. *"Model soups: averaging weights of multiple fine-tuned models improves accuracy without increasing inference time."* International Conference on Machine Learning (ICML), PMLR, 2022.
> 2. Khirodkar, Rawal, et al. *"Sequential Ensembling for Semantic Segmentation."* arXiv preprint arXiv:2210.05387, 2022.

---

> ### Author Response · Authors · 2024-11-27
> **Follow-Up on Our Response to Your Feedback**
>
> Dear Reviewer pQF1,
>
>
> Thank you once again for your thoughtful and detailed review of our paper. We have carefully addressed all of your feedback and incorporated the suggested changes into the manuscript.
>
> If there are any additional questions, concerns, or suggestions, we would be more than happy to address them promptly. Please do not hesitate to share any further input or clarification you may require. We remain committed to ensuring that our responses align with your expectations and hope they have satisfactorily resolved the points you raised.
>
> Thank you for your time and consideration, and we look forward to hearing from you at your earliest convenience.
>
>
> Sincerely,
> The Authors

---

> ### Comment · Reviewer_pQF1 · 2024-11-29
> **Response to feedback 2/8**
>
> The limitations and challenges discussed by the authors are primarily based on performance evaluations within seen datasets, which still fall under the closed-set category and do not address open-vocabulary scenarios.
>
> The authors state, "We demonstrated that OVS models are effective only within the domains of their training datasets." All models tend to perform poorly when encountering datasets that were not seen during training, regardless of whether they are OVS models. For instance, in Table 1, Mask2Former exhibits high performance on Cityscapes and ADE20K because it was trained on these datasets, whereas X-Decoder and FC-Clip are evaluated in a zero-shot manner on the same datasets, resulting in lower performance.
>
> The authors also mention that fine-tuning OVS models on new datasets leads to catastrophic forgetting, significantly degrading performance on previously learned tasks. However, these datasets are all seen datasets and still belong to the closed-set category. The critical aspect of OVS is its performance on unseen datasets, and I do not see specific elements in your approach that are explicitly designed for OVS.

---

> > ### Author Response · Authors · 2024-11-30
> > **Response to Additional Feedback from Reviewer pQF1 (1/3)**
> >
> > We sincerely appreciate the reviewer's thoughtful follow-up feedback and the opportunity to provide further clarification. Below, we address the reviewer's questions in detail.
> >
> > **Q1) The limitations and challenges discussed by the authors are primarily based on performance evaluations within seen datasets, which still fall under the closed-set category and do not address open-vocabulary scenarios.**
> >
> > During the Rebuttal period, we recognized, based on the reviewers' comments, that our evaluation of the model’s performance as an OVS model was insufficient. To address this, we conducted additional performance evaluations on unseen datasets and incorporated these results into the tables. Below, we provide responses to each question along with detailed explanations of the improvements made.
> >
> > **Q2) The authors state, "We demonstrated that OVS models are effective only within the domains of their training datasets." All models tend to perform poorly when encountering datasets that were not seen during training, regardless of whether they are OVS models. For instance, in Table 1, Mask2Former exhibits high performance on Cityscapes and ADE20K because it was trained on these datasets, whereas X-Decoder and FC-Clip are evaluated in a zero-shot manner on the same datasets, resulting in lower performance.**
> >
> > That is correct—the exact point raised by the reviewer aligns with what we aimed to convey. Whether it is an OVS model or not, models often fail to perform well on datasets unseen during training. This is precisely what we intended to highlight, as OVS models are generally believed to work effectively even on unseen datasets. Our objective was to challenge this assumption by demonstrating that OVS models may not actually fulfill this expectation.
> >
> > In Table 1, we aimed to show that state-of-the-art OVS models perform poorly even on data distributions closely related to real-life scenarios, such as driving scenes. While fine-tuning on these datasets may appear to resolve this issue, as shown in Figure 1, fine-tuning leads to a substantial performance drop on other data distributions. This implies that while fine-tuning improves performance on the specific dataset, it compromises the essential characteristics of an OVS model. Since no prior work has addressed this issue with OVS models, we used Table 1 and Figure 1 to highlight this problem and demonstrate that methods like fine-tuning cannot resolve it.
> >
> > Subsequently, in Table 2, we showed that even in the most ideal training environment, such as joint training, this issue persists. Furthermore, we demonstrated that continual learning methods, which are designed to mitigate catastrophic forgetting, also fail to address the problem. This highlights the need for new approaches that can effectively expand the range of data distributions OVS models can handle while preserving their unique properties—particularly their ability to perform well on unseen data distributions.
> >
> > We acknowledge that this point was not clearly articulated in the original manuscript. To address this, we have completely rewritten the relevant paragraphs (Lines 39-53, Lines 220-235). Additionally, we incorporated joint training as a baseline in the key performance comparison tables (Table 2, Table 4, Table 5) and included performance results on unseen datasets to more thoroughly demonstrate the capabilities of OVS models (Table 2, Table 3, Table 5).
> >
> > We hope this revision clarifies our message and addresses the reviewer’s concerns. If you have further questions, please feel free to let us know.
> >
> > **Q3) The authors also mention that fine-tuning OVS models on new datasets leads to catastrophic forgetting, significantly degrading performance on previously learned tasks. However, these datasets are all seen datasets and still belong to the closed-set category. The critical aspect of OVS is its performance on unseen datasets, and I do not see specific elements in your approach that are explicitly designed for OVS.**
> >
> > We agree with the reviewer’s perspective on the importance of evaluating the performance of OVS models on unseen datasets. As mentioned earlier, during the rebuttal period, we incorporated the reviewers' feedback by adding performance results on unseen datasets to the main tables and actively utilizing them in our analysis.
> >
> > Our experimental results show that fine-tuning an OVS model leads to a performance decrease not only on previously trained data distributions but also on unseen datasets. In contrast, our method preserves the OVS model’s inherent capability to perform well on unseen datasets while significantly improving performance on the fine-tuning dataset. This effectively expands the range of data distributions that the OVS model can handle. We have demonstrated this clearly in **Figure 1** and **Tables 2, 3, and 5**, and further emphasized it in the manuscript.
> >
> > We hope this response addresses the reviewer’s concerns. Please feel free to reach out with any additional questions.

---

### Official Review · Reviewer_bfmq · 2024-10-30

**Soundness:** 3
**Presentation:** 3
**Contribution:** 3
**Rating:** 5
**Confidence:** 2

**Summary:**

The paper addresses the domain limitations in Open-Vocabulary Segmentation models, which tend to perform poorly on unseen domains and suffer from catastrophic forgetting when fine-tuned on new datasets. This paper proposes to use precomputed multivariate normal distributions for domain representation and adjusts the model weights during inference, allowing the OVS models to adapt to new domains while retaining performance on previously trained datasets.

**Strengths:**

The method introduced in the paper enhances the adaptability of Open-Vocabulary Segmentation models to new domains without the need for retraining from scratch or accessing previous datasets during training. It effectively addresses the problem of catastrophic forgetting by maintaining consistent performance on previously trained datasets while adapting to new domains. Additionally, the method does not introduce additional parameters into the OVS model, allowing for seamless integration with existing architectures and ensuring computational efficiency.

**Weaknesses:**

The paper presents an innovative approach to handling domain limitations in Open-Vocabulary Segmentation (OVS) models, but there are several considerations that may impact its practical deployment:

Challenges in Implementation: This method involves calculating interpolation coefficients and blending multiple decoders into one during inference, which could require substantial computational resources. It might be complex to tune and stabilize, possibly complicating its use in real-world scenarios. Have there been any quantitative analyses that compare the computational demands (like time and operations) of this method against others during inference?


Use of Multiple Evaluation Metrics: Why do Tables 4 and 5 use a mix of mAP, PQ, and mIoU metrics for evaluation? Understanding the rationale behind choosing these diverse metrics could help clarify their relevance to the goals of the study.

Managing Computational Costs: Are there existing methods or technologies that could be adapted to help reduce the computational costs associated with this method? Exploring how similar challenges have been addressed in other contexts might provide viable solutions to enhance the efficiency of this approach.

**Questions:**

See weaknesses

---

> ### Author Response · Authors · 2024-11-23
> **Response to Reviewer bfmq (1/4)**
>
> Thank you for recognizing the strengths of our method, particularly its ability to enhance adaptability to new domains while addressing catastrophic forgetting without the need for retraining or additional parameters. We appreciate your acknowledgment of its seamless integration and computational efficiency, as these aspects were central to our design goals. Below, we address your concerns.

---

> ### Author Response · Authors · 2024-11-23
> **Response to Reviewer bfmq (2/4)**
>
> ### **Q1) Computational Challenges in Implementation**
>
> *This method involves calculating interpolation coefficients and blending multiple decoders into one during inference, which could require substantial computational resources. It might be complex to tune and stabilize, possibly complicating its use in real-world scenarios. Have there been any quantitative analyses that compare the computational demands (like time and operations) of this method against others during inference?*
>
> ---
>
> As the reviewer correctly pointed out, our method requires additional computational resources, but the increase in overall inference time remains minimal. Unlike other continual learning techniques, our approach does not require additional parameters as the number of learned datasets grows [1], nor does it involve multiple forward passes [2]. Moreover, by applying weight interpolation exclusively to the decoder in encoder-decoder models, our method ensures computational efficiency.
>
> #### **Inference Time Analysis**
>
> | **Number of Seen Datasets** | **Encoder (ms)** | **Interpolation Factor Estimator (ms)** | **Decoder Weight Interpolation (ms)** | **Decoder (ms)** | **Total Inference Time Per Sample (ms)** | **Change (%)** |
> |-----------------------------|------------------|-----------------------------------------|---------------------------------------|------------------|-------------------------------------------|----------------|
> | 1                           | 97.69           | -                                       | -                                     | 102.30           | 199.99                                   | +0.00%       |
> | 2                           | 97.69           | 0.81                                    | 10.69                                 | 102.30           | 211.48                                   | +5.75%      |
> | 3                           | 97.69           | 1.01                                    | 13.23                                 | 102.30           | 214.23                                   | +7.12%       |
>
> Based on these results, training with two datasets led to a 5.75 percentage point increase in inference time compared to the baseline, while adding an additional dataset resulted in only a 1.37 percentage point increase. These findings indicate that the overall impact on inference time is negligible compared to the inference times of the encoder and decoder.
>
> #### **Comparison with PEFT [1]**
>
> | **# Seen Datasets** | **PEFT [1] (ms)**      | **Ours (ms)**        |
> |---------------------|------------------------|----------------------|
> | 1                   | 199.99 (+0.00%)       | 199.99 (+0.00%)     |
> | 2                   | +36.04 (+18.02%)      | +11.49 (+5.75%)     |
> | 3                   | +84.73 (+42.37%)      | +14.24 (+7.12%)     |
>
> We also compared the inference time of our method with an existing continual learning technique [1] designed for closed-set segmentation. Using the COCO validation set, we measured the average inference time per sample, as detailed in the table above. Our method demonstrated faster inference times than PEFT [1] and showed smaller increases in inference time as the number of seen datasets grew.
>
> #### **Storage Demands**
>
> Our method requires saving the parameters of the learned decoders for each dataset. However, since only the decoder parameters are stored (rather than the entire model), the storage demand is significantly lower compared to other model ensemble approaches [3, 4].
>
> Specifically, our method stores only *6.11% of the total model size* (approximately 80MB), which greatly reduces the storage requirements for model merging. This ensures that our approach does not impose a substantial storage overhead relative to the size of the full model. We have included this discussion in the revised manuscript.
>
> ---
>
> #### **Conclusion**
>
> In conclusion, we believe the relatively low impact of our method on inference time makes it practical for real-world applications. That said, we acknowledge that our earlier work lacked sufficient analysis of computational demands. To address this, we have included additional inference time experiments and analyses in **Section 5.3**. Thank you for raising this important point.
>
> ---
>
> #### **References**
>
> [1] Kim, Beomyoung, Joonsang Yu, and Sung Ju Hwang. "ECLIPSE: Efficient Continual Learning in Panoptic Segmentation with Visual Prompt Tuning.", In CVPR, 2024.
> [2] Wang, Yabin, Zhiwu Huang, and Xiaopeng Hong. "S-prompts learning with pre-trained transformers: An Occam’s razor for domain incremental learning.", In NeurIPS, 2022.
> [3] Wortsman, Mitchell, et al. *"Model soups: averaging weights of multiple fine-tuned models improves accuracy without increasing inference time."* International Conference on Machine Learning (ICML), PMLR, 2022.
>
> [4] Khirodkar, Rawal, et al. *"Sequential Ensembling for Semantic Segmentation."* arXiv preprint arXiv:2210.05387, 2022.

---

> ### Author Response · Authors · 2024-11-23
> **Response to Reviewer bfmq (3/4)**
>
> ### **Q2) Use of Multiple Evaluation Metrics**
>
> *Why do Tables 4 and 5 use a mix of mAP, PQ, and mIoU metrics for evaluation? Understanding the rationale behind choosing these diverse metrics could help clarify their relevance to the goals of the study.*
>
> ---
>
> The reason for using different evaluation metrics in Tables 4 and 5 is that the segmentation tasks vary across the datasets. We evaluated the models using the most representative metric suitable for each task:
>
> - **Semantic Segmentation Tasks:** For the Mapillary, Pascal VOC, Pascal Context, and ADE20k-847 datasets, we used *mIoU*, as these datasets are designed for semantic segmentation tasks.
> - **Panoptic Segmentation Tasks:** For BDD100K, we used *PQ*, since this dataset targets panoptic segmentation.
> - **Instance Segmentation Tasks:** For LVIS, we employed *mAP*, as it focuses on instance segmentation tasks.
>
> We have added a clarification in Appendix B to explain the rationale for selecting different evaluation metrics based on the datasets. Thank you for bringing this to our attention.

---

> ### Author Response · Authors · 2024-11-23
> **Response to Reviewer bfmq (4/4)**
>
> ### **Q3) Managing Computational Costs**
>
> *Are there existing methods or technologies that could be adapted to help reduce the computational costs associated with this method? Exploring how similar challenges have been addressed in other contexts might provide viable solutions to enhance the efficiency of this approach.*
>
> ---
>
> Our method introduces a computational overhead in two key stages:
> 1. **Estimating the interpolation factor:** This stage has a minimal impact on the overall inference time, as previously mentioned.
> 2. **Performing weight interpolation:** This stage is where we focused on exploring techniques to reduce computational costs.
>
> #### **Techniques to Reduce Computational Costs**
>
> One prominent approach to reducing the computational cost of weight interpolation is to **minimize the number of parameters involved in the merging process**. This method has been actively studied in the model merging field. Examples include:
>
> - **Pruning Redundant Parameters:** Several works [1,2,3,4] propose identifying and pruning redundant parameters across multiple models before merging. This enhances efficiency while minimizing interference between models.
> - **Mixture-of-Experts Approaches:** Another example is training only a subset of parameters per task and merging them afterward, as demonstrated in [5].
>
> #### **Challenges and Future Directions**
>
> Applying these model merging approaches directly to segmentation models, particularly OVS models, remains challenging due to the lack of prior studies in this area. Nonetheless, reducing inference time for OVS models is a crucial problem. Exploring new methods for efficient weight interpolation offers a promising avenue for future work.
>
> This discussion has been included in **Section 6 (Limitations)** of the paper. Thank you for the constructive question.
>
> ---
>
> #### **References**
>
> [1] Yadav, Prateek, et al. "Ties-merging: Resolving interference when merging models." *Advances in Neural Information Processing Systems* 36 (2024).
> [2] Deep, Pala Tej, Rishabh Bhardwaj, and Soujanya Poria. "DELLA-Merging: Reducing Interference in Model Merging through Magnitude-Based Sampling." *arXiv preprint arXiv:2406.11617* (2024).
> [3] Sun, Mingjie, et al. "A simple and effective pruning approach for large language models." In *ICLR, 2024*.
> [4] Zimmer, Max, Christoph Spiegel, and Sebastian Pokutta. "Sparse model soups: A recipe for improved pruning via model averaging." *arXiv preprint arXiv:2306.16788* (2023).
> [5] Tang, Anke, et al. "Merging Multi-Task Models via Weight-Ensembling Mixture of Experts." *arXiv preprint arXiv:2402.00433* (2024).

---

> ### Author Response · Authors · 2024-11-27
> **Follow-Up on Our Response to Your Feedback**
>
> Dear Reviewer bfmq,
>
> Thank you once again for your thoughtful and detailed review of our paper. We have carefully addressed all of your feedback and incorporated the suggested changes into the manuscript.
>
> If there are any additional questions, concerns, or suggestions, we would be more than happy to address them promptly. Please do not hesitate to share any further input or clarification you may require. We remain committed to ensuring that our responses align with your expectations and hope they have satisfactorily resolved the points you raised.
>
> Thank you for your time and consideration, and we look forward to hearing from you at your earliest convenience.
>
> Sincerely,
> The Authors

---

> ### Author Response · Authors · 2024-12-01
> **Kind Follow-Up on Discussion Period Feedback**
>
> Dear Reviewer,
>
> With only two days left in the discussion period, we kindly follow up on our earlier reminders regarding your feedback. We have carefully worked to address the questions and concerns you raised through our responses and revisions, and we believe that we have resolved most, if not all, of the issues you highlighted.
>
> If our efforts and revisions meet your expectations, we would greatly value your acknowledgment. However, if you have any additional concerns, we are more than willing to address them promptly before the discussion period concludes.
>
> Thank you for your time and thoughtful consideration.
>
> Sincerely,
> The Authors

---

### Official Review · Reviewer_mCxt · 2024-11-01

**Soundness:** 2
**Presentation:** 2
**Contribution:** 2
**Rating:** 3
**Confidence:** 1

**Summary:**

The paper proposes to combine continual learning and open vocabulary image segmentation in order to improve the generalizability of the model. Its motivation is that this paper finds that finetuning and simply applying continual learning can degrade the performance on unseen new datasets and so proposes to use MVN distributions to adapt to new data. Specially, it saves the means and covariance matrices of the image and test embeddings for each seen dataset. When the new dataset arrives, just fit it to the distribution with an interpolation factor vector. The experiment is based on two popular models, fc-clip and X-Decoder, and shows improvement over the finetuning method.

**Strengths:**

The proposed method is easy to understand and looks like easy to implement.

The experiment and ablation study on fc-clip is sufficient.

**Weaknesses:**

The presentation is weak and the expression is not rigorous. For example, the definition of "domain" in this paper is unclear. The problem setting also needs more clarification. Some information details are missing.

Many necessary experiments are missing.

For the proposed method, significant effort is required to assess its soundness. For example, is it reasonable to construct an MVN distribution based on different datasets? for coco2014 and 2017, do we need to treat them to two distinct datasets?

For details of all points, please refer to my listed questions.

**Questions:**

For table, what does exactly domain mean? I think that is just cross dataset validation, not cross-domain. For different domains, that usually means images with different styles like DomainNet datasets or from synthetic images to real world images. And the comparison in table 1 is unfair. I think for Mask2Former, that is the fine-tuned number. And Since the architecture is also different from Mask2Former and X-Decoder and fc-clip. For a better comparison, why not training X-Decoder on Cityscapes and ADE20K, instead of using Mask2Former? Also for fc-clip.

Line 43, why do we need fine-tuning. these years, many models have been universal. They can be trained on different datasets jointly. In that case, we don’t need finetuning and therefore there is no catastrophic forgetting problem.

Line 54-55, which methods? Reference?

Line 57, needs to clarify, what does “Our method begins by fine-tuning the decoder” mean? For open-vocabulary setting, should we assume know the domain of the new dataset? If not, why can we fine-tune on it?

Figure 1 is unclear. What do “previous”, “fine-tuning”, “unseen” mean? Please use professional terms, source domain, target domain to clarify. From my understanding, I think previous is the source domain and unseen is the target domain, if so, what is fine-tuning? Also, the bar chart is for what model? At least should be pointed in somewhere in the experiment section.

Line 89, “Despite these advancements, current methods remain effective only within the domain of the previous training dataset and fail to generalize to new domains.”, please tone down the expression, if current methods fail to generalize to new domains. Why too many open-vocabulary segmentation works appeared. For example, OpenSeeD can work well on SeginW dataset that has multiple subsets of different domains (just use the word in this paper). Why it fails.

Line 123, again, for Figure 1, what is the model?

Line 134, “with each batch containing an equal proportion of data from both datasets.” Any reference? I don’t think it has to be equal proportion of data from both datasets. Sometimes the ratio depends on the quality of the mixed datasets.

Line 137, for the second issue “2) Whether the model is trained from scratch or fine-tuned on both datasets, joint training consistently results in lower performance compared to fine-tuning on the target dataset alone (see Figure 2).”. I can agree. It is unclear what the used model is. But here COCO to Cityscapes is just one specific example. If jointly training is better or worse than fine-tuning depends on the domain gap (still use the word in this paper, although I think use dataset gap is more accurate). For example, is the performance of jointly training on COCO and LVIS is better than that of finetuning on COCO?

For the third issue, why not using downsampling and upsampling to control that.

For the first issue, why it is challenging? Is it hard to just consume some storage to save some data? Any big difference from storing means and covariance matrices for each datasets in this paper?

Line 157, what does PEFT mean?

Can’t understand Figure 3 (b), what does “average inference time per sample” mean? Why the inference time varies according to the Number of seen datasets? What does “Number of seen datasets” mean here? Source datasets + newly arrived datasets?, shouldn’t the inference time only depends on the model, testing sample and device? What these three datasets are?

I think the problem definition is not open-vocabulary. For OV, the testing images and class is unknown, but in this paper, it allows sequential fine-tuning. So I think the paper is just a continual learning of segmentation.

For the algorithm 1, I don’t think it makes much sense to save means and covariance matrices for each datasets. If it makes sense? Why not doing that for each subgroups of these datasets? We can’t guarantee the domain gap of two images across datasets is certainly larger than two images inner the same dataset. So a more reasonable way is to cluster all seen data and proceed the algorithm for each cluster.

Overall, the MVN distribution is similar to those typical multi domain adaptation methods, For example, “Li, Yunsheng, Lu Yuan, Yinpeng Chen, Pei Wang, and Nuno Vasconcelos. "Dynamic transfer for multi-source domain adaptation." In CVPR.” In the sense that to compute an interpolation factor vector and like a linear combination of source domains in order to adapt to target domain. But anyway, I think in the section 2,  multi domain adaptation literature review is recommended.

For experiments, I think a critical baseline missing is jointly training on all previous and fine-tuning datasets.

Again, why not comparing to other OVS methods like ODISE, OpenSeeD, Open-vocabulary SAM, etc? The setting is easy, just use all previous and fine-tuning datasets to train their models and evaluate on these eight unseen datasets. Otherwise, the results shown like in tables 3, 4 make the reader hard to believe the proposed method work.

Another big problem is the experiments need cross validation, not just usually COCO as the previous training set. In real world, we could start with ADE, we could start with Cityscapes, or we could also start with a subset like 10K training samples of COCO. In these cases, does the proposed method still work well?

---

> ### Author Response · Authors · 2024-11-20
> **Response to Reviewer mCxt (1/10)**
>
> Thank you for your thoughtful review and valuable feedback. Below we address your concerns.

---

> ### Author Response · Authors · 2024-11-20
> **Response to Reviewer mCxt (2/10)**
>
> **Q1) What does the term "domain" mean in this paper?**
>
> As the reviewer pointed out, the term "domain" as defined in this paper differs from its usage in conventional domain adaptation tasks. In this paper, we refer to the data distribution that each dataset follows. For example, while both COCO and Cityscapes consist of real-world photographic images, COCO represents a general data distribution, whereas Cityscapes follows a specific data distribution tailored for drivers. To avoid confusion, we have replaced the term "domain" with "data distribution" throughout the revised manuscript. Thank you for the insightful feedback.
>
> ---
>
> **Q2) The comparison in Table 1 seems unfair since Mask2Former is fine-tuned. Why not fine-tune X-Decoder and fc-CLIP on Cityscapes and ADE20K for a fairer comparison?**
>
> As the reviewer noted, it is indeed unfair to compare fine-tuned Mask2Former models with open-vocabulary segmentation (OVS) models that are not fine-tuned. However, this comparison was intentional. Specifically, we aimed to demonstrate that conventional segmentation models fine-tuned on a specific dataset significantly outperform OVS models. The key point we wanted to highlight is that OVS models should ideally perform well on unseen datasets, but in practice, they often fail to do so. If OVS models perform worse than closed-set segmentation models, there would be little incentive to use them.
>
> That said, the reviewer's comment made us realize the value of conducting a fair comparison as well. To address this, we fine-tuned X-Decoder and fc-CLIP on Cityscapes and ADE20K, respectively. The results of these experiments are shown in the table below.
>
> | **Method**      | **Fine-tuning** | **Cityscapes** | **ADE20k** |
> |------------------|-----------------|----------------|------------|
> | Mask2Former      | O               | 62.1           | 39.7       |
> | X-Decoder        | X               | 36.2           | 16.7       |
> | X-Decoder        | O               | **62.9**       | **44.9**   |
> | fc-CLIP          | X               | 44.0           | 26.8       |
> | fc-CLIP          | O               | **64.2**       | **47.6**   |
>
> These findings further reinforce our original message: *"OVS models perform poorly unless fine-tuned on the target dataset."* We appreciate the constructive feedback and have included this discussion and the updated table in Section 1.
>
> ---
>
> **Q3) Line 43: Why is fine-tuning needed? Many recent models are universal and trained on multiple datasets jointly, avoiding the need for fine-tuning and catastrophic forgetting.**
>
> We believe that leveraging all available datasets for joint training is not a scalable solution. As shown in Table 1, open-vocabulary segmentation models like X-Decoder and fc-CLIP exhibit significantly lower performance on datasets not used for their training. CityScapes and ADE20K are datasets that represent common, everyday scenarios, and therefore, such performance is less than ideal. Recognizing this limitation, we identified a need for methods to customize pre-trained SOTA OVS models to improve performance on specific datasets while preserving their open-vocabulary recognition capabilities. Our approach provides a solution to this demand.
>
> We hope this addresses the reviewer’s concern, and if not, we would appreciate further feedback.
>
> ---
>
> **Q4) Lines 54–55: Could you clarify which specific methods are being referred to?**
>
> As the reviewer pointed out, we noticed that references were missing in Lines 54–55. To address this, we have cited the following two papers in the revised manuscript. Thank you for the constructive feedback.
>
> 1. Kirkpatrick, James, et al. "Overcoming catastrophic forgetting in neural networks." *Proceedings of the National Academy of Sciences* 114.13 (2017): 3521-3526.
> 2. Kim, Beomyoung, Joonsang Yu, and Sung Ju Hwang. "ECLIPSE: Efficient Continual Learning in Panoptic Segmentation with Visual Prompt Tuning." *CVPR 2024*.
>
> ---
>
> **Q5) Line 57: What does “Our method begins by fine-tuning the decoder” mean? Does this assume the new dataset's domain is known?**
>
> As mentioned in Q3, our work is based on the assumption of a scenario where a pre-trained OVS model is required to enhance its performance on a user-specified dataset without compromising its open-vocabulary recognition capabilities. This scenario inherently assumes that the new dataset is already known to the user. We believe this is a practical and relevant assumption, as it aligns with real-world applications where users seek to adapt existing models to specific tasks or datasets. However, we acknowledge that this assumption was not explicitly articulated in the current version of the manuscript. To address this, we have revised the manuscript to include a detailed clarification of this assumption, inserting it before the sentence at Line 57 for better contextual understanding.

---

> > ### Comment · Reviewer_mCxt · 2024-11-27
> > **thanks for the reply.**
> >
> > The section highlighted in lines 44 to 50 appears to conflict with the values presented in Table 1. Specifically, 62.9/64.2 is greater than 62.1, and 44.9/47.6 exceeds 39.7. Why, then, does the text state that these models 'perform worse'?
> > Additionally, if the assumption is that the target new dataset is already known, then comparing to the current OVS models fc-clip/x-decoder is unfair, as the latter does not operate under this assumption. Also, table 2 is somewhat unfair. In this sense, the proposed method is more like an unsupervised domain adaption setting. Not open-vocabulary.

---

> > > ### Author Response · Authors · 2024-11-28
> > > **Response to Additional Feedback from Reviewer mCxt (1/3)**
> > >
> > > We sincerely appreciate your thoughtful follow-up feedback and the opportunity to further clarify our responses. Before addressing the reviewer’s question, we would like to first outline the scenario we aim to address and the contributions of our paper.
> > >
> > > We propose a novel approach to adapt open-vocabulary segmentation (OVS) models to perform effectively in specific data distributions while preserving their inherent open-vocabulary capabilities. The scenario we consider is as follows: imagine a user intends to deploy an OVS model in a specific environment, such as driving scenes. In such cases, pre-trained OVS models often fail to deliver satisfactory performance on the unseen target data distribution. Recognizing this gap, we identified the need for a solution that retains the original capabilities of the OVS model while also adapting it to perform well in environments like driving scenes. This represents a practical challenge, especially for small-scale enterprises or research labs with limited computational resources.
> > >
> > > Fine-tuning may appear as a straightforward solution to this issue. However, fine-tuning often leads to the loss of the model’s open-vocabulary characteristics. We also explored approaches such as joint training and continual learning as potential solutions to this scenario. However, our analysis revealed that these approaches fall short in addressing the problem effectively. To overcome this limitation, we propose a method that adapts OVS models to align with specific data distributions while preserving their open-vocabulary capabilities. This enables users to achieve high performance in desired environments without compromising the foundational advantages of OVS models.
> > >
> > > We now proceed to address the reviewer’s specific question below.
> > >
> > >
> > > **Q1) The section highlighted in lines 44 to 50 appears to conflict with the values presented in Table 1. Specifically, 62.9/64.2 is greater than 62.1, and 44.9/47.6 exceeds 39.7. Why, then, does the text state that these models 'perform worse'?**
> > >
> > > The key point we aim to convey is that fine-tuning is necessary to make OVS models surpass the performance of closed-set segmentation models on a given data distribution. As shown in Table 1, OVS models without fine-tuning exhibit lower performance compared to closed-set segmentation models (e.g., on Cityscapes, fc-clip: 36.2, X-Decoder: 44.0 vs. Mask2Former: 62.1). In contrast, applying fine-tuning to OVS models allows them to achieve superior performance (e.g., on Cityscapes, fc-clip: 62.9, X-Decoder: 64.9 vs. Mask2Former: 62.1). We recognize that the previous description was ambiguous, and to address this, we propose revising the paragraph as follows (Line 37-52):
> > >
> > > > Despite these advancements, we observe that OVS models are effective only within the data distribution of their source datasets. As shown in Table 1, OVS models perform significantly worse than closed-set segmentation models when evaluated on datasets with a different data distribution from the source dataset. Fine-tuning OVS models on these datasets can lead to substantial performance improvements; however, as illustrated in Figure 1, this comes at the cost of a significant drop in performance on unseen target data distributions. This limitation greatly restricts the applicability of OVS models in scenarios where recognizing objects in unseen target data distributions is critical.

---

> > > ### Author Response · Authors · 2024-11-28
> > > **Response to Additional Feedback from Reviewer mCxt (2/3)**
> > >
> > > **Q2) If the assumption is that the target new dataset is already known, then comparing to the current OVS models (fc-clip/x-decoder) is unfair, as the latter does not operate under this assumption. Also, Table 2 is somewhat unfair. In this sense, the proposed method is more like an unsupervised domain adaptation setting, not open-vocabulary.**
> > >
> > > First, it is indeed true that existing OVS models were developed under the assumption that they are unaware of the target dataset. However, our analysis indicates that OVS models do not always perform reliably across all target datasets. As we mentioned above, there are scenarios where achieving high performance on a specific dataset is critical. Our study provides a solution for such a scenario, which allows the model to efficiently expand its coverage of the data distribution it can handle.
> > >
> > > Next, regarding the concern that Table 2 might be unfair, we would like to clarify its intent. This table was designed not to compare performance with existing OVS techniques but to benchmark against potential solutions under the same assumptions and with the same dataset. Table 2 demonstrates that straightforward solutions like joint training and fine-tuning do not perform effectively in our scenario. Additionally, it highlights that existing continual learning methods also struggle with the proposed problem. Hence, we believe Table 2 provides a fair comparison.
> > >
> > > In response to the question regarding the alignment of our approach with UDA, we emphasize that our study addresses an open-vocabulary task, which is fundamentally distinct from UDA. In our task, the target dataset contains classes that differ from those in the source dataset, while UDA typically assumes shared classes between domains. Moreover, our approach prohibits the use of the source dataset during training, further distinguishing it from UDA.
> > >
> > > Additionally, UDA assumes prior knowledge of the target data distribution, whereas our method operates without this information. Instead, we only utilize the fine-tuning dataset, and the target data distribution remains unknown, highlighting a key difference. To assess performance, we evaluate our method on multiple target datasets. For instance, Tables 5 and 7 present experiments across various types of target datasets.
> > >
> > > That said, we acknowledge the reviewer's concern that reporting results on only a single target dataset in Table 2 does not adequately evaluate the model's capability as an OVS model. To address this, we have updated Table 2 to include a new column reporting the average performance across six target datasets, providing a more comprehensive assessment.
> > >
> > > **(a) Cityscapes**
> > >
> > > |Method|COCO (source)|Cityscapes (fine-tuning)|ADE20K (target)|Avg on 6 target datasets|
> > > |------|------------|------------------------|---------------|------------------------|
> > > |fc-clip|50.1|44.0|23.5|45.6|
> > > |+ Fine-tuning|-22.7|+20.1|-10.3|-3.9|
> > > |+ Joint Training|+0.6|+17.9|+1.7|+0.5|
> > > |+ ER|-1.6|+19.0|+0.3|-0.6|
> > > |+ LwF|-10.7|+12.2|-0.8|-1.1|
> > > |+ EWC|-25.9|+19.3|-9.8|-4.3|
> > > |+ ECLIPSE|-6.0|+2.2|+0.9|-0.7|
> > > |+ Ours|**+0.3**|**+20.2**|**+2.5**|**+0.6**|
> > >
> > > **(b) ADE20K**
> > >
> > > |Method|COCO (source)|Cityscapes (fine-tuning)|ADE20K (target)|Avg on 6 target datasets|
> > > |------|------------|------------------------|---------------|------------------------|
> > > |fc-clip|50.1|23.5|44.0|46.0|
> > > |+ Fine-tuning|-7.7|+24.1|-3.0|+0.3|
> > > |+ Joint Training|+1.4|+16.5|-1.2|+0.6|
> > > |+ ER|+0.4|+21.5|-3.5|+0.0|
> > > |+ LwF|-3.8|+13.7|-1.0|-0.4|
> > > |+ EWC|-11.1|+20.7|-2.6|-1.5|
> > > |+ ECLIPSE|-0.5|+0.2|-5.9|-0.4|
> > > |+ Ours|**+1.7**|**+23.8**|**-0.3**|**+0.6**|
> > >
> > > Our ultimate goal is to enhance performance on the fine-tuning dataset while maintaining the characteristics of an OVS model. While existing approaches such as joint training, fine-tuning, and continual learning fail to achieve this, our proposed method successfully does so, as shown in our results. We hope this clarification addresses your concerns.

---

> > ### Comment · Reviewer_mCxt · 2024-11-27
> > **confusion about the problem definition**
> >
> > I just re-read the revised problem definition section, lines 224-234. It is still unclear. Line 230, what is exactly ‘seen’ datasets, source datasets + a bunch of fine tuned datasets? If so, this conflicts with the statement in the paper and rebuttal where for example, it states that in the i-th fine-tuning stage, the model has access only to the current dataset, D^i_ft. Why can't other D^i_ft sometimes be accessed but during evaluation, it is suddenly accessible. And I don’t understand this “The main challenge is to improve performance on the new dataset D^i_ft? D^i_ft is a new dataset? Shouldn’t it is the fine-tuned datasets as the stated. It has been used during training?

---

> ### Author Response · Authors · 2024-11-20
> **Response to Reviewer mCxt (3/10)**
>
> **Q6) Could you clarify the terms "previous," "fine-tuning," and "unseen"?**
>
> As the reviewer pointed out, we recognize that the terms we used are not formal. Following the reviewer's suggestion, we have replaced *"previous"* with *"source"* and *"unseen"* with *"target."* The term *"previous"* was intended to refer to datasets previously used for training, which aligns exactly with the meaning of *"source."* Similarly, *"unseen"* refers to datasets not used during training, where performance is critical for evaluating the open-vocabulary capabilities of the model. Therefore, *"target"* is indeed a more appropriate term. On the other hand, we believe the term *"fine-tuning dataset"* clearly conveys its intended meaning—datasets used to fine-tune the OVS model—and thus we have not changed it. We apologize for any confusion caused by our terminology choices and hope this revision addresses the reviewer's concern. If you have further suggestions, we would be happy to consider them.
>
> ---
>
> **Q7) In Figure 1 and 2, what is the OVS model used?**
>
> We acknowledge that Figures 1 and 2 both utilize fc-CLIP, and this information was not explicitly stated in the manuscript. We have updated the figure captions to include this detail. Thank you for the constructive question.
>
> ---
>
> **Q8) Line 89: The statement “fail to generalize to new domains” seems too strong. Please tone down the expression.**
>
> We recognize that the phrase *“fail to generalize to a new domain”* is an overstatement, especially given the ongoing advancements in OVS models and their demonstrated success in certain scenarios. Taking this into account, we have revised it to state that *“current OVS models, when not trained on specific datasets, can exhibit significantly lower performance.”* We believe this revision clearly conveys the limitations of current OVS models.
>
> ---
>
> **Q9) Line 134: “with each batch containing an equal proportion of data from both datasets.” Could you provide a reference for this? I believe it doesn’t necessarily have to be an equal proportion of data from both datasets.**
>
> To ensure balanced learning across multiple datasets, we adopted a straightforward approach of uniformly sampling within each batch during joint training. This method is commonly used in prior studies to balance data in a batch based on specific criteria when training on multiple datasets or multi-modal datasets. For example, Experience Replay methods [1, 2] in continual learning typically balance the ratio of previous and new datasets within a batch. Similarly, [3] proposes balancing different modalities by uniformly including various modality data within a batch during multi-modal joint training.
>
> However, as the reviewer mentioned, batch composition can be adjusted based on dataset quality, but this approach introduces additional costs compared to uniformly sampling across all datasets. For instance, determining dataset quality through a network incurs extra training costs [4, 5], while manual assessment of sample quality requires additional annotation costs [6].
>
> Given these considerations, we applied a uniform composition of samples from the two datasets within each batch during joint training. We have included an explanation of this approach and add the relevant references in Section 3.2.
>
> References:
>
> 1. Rolnick, David, et al. "Experience replay for continual learning." *Advances in Neural Information Processing Systems* 32 (2019).
> 2. Van de Ven, Gido M., Tinne Tuytelaars, and Andreas S. Tolias. "Three types of incremental learning." *Nature Machine Intelligence* 4.12 (2022): 1185-1197.
> 3. Evans, Talfan, et al. "Data curation via joint example selection further accelerates multimodal learning." *arXiv preprint arXiv:2406.17711* (2024).
> 4. Yoon, Jinsung, Sercan Arik, and Tomas Pfister. "Data valuation using reinforcement learning." *International Conference on Machine Learning*. PMLR, 2020.
> 5. Jia, Ruoxi, et al. "Towards efficient data valuation based on the Shapley value." *AISTATS 2019*. PMLR, 2019.
> 6. Fleckenstein, Mike, Ali Obaidi, and Nektaria Tryfona. "Data Valuation: Use Cases, Desiderata, and Approaches." *Proceedings of the Second ACM Data Economy Workshop*. 2023.

---

> ### Author Response · Authors · 2024-11-20
> **Response to Reviewer mCxt (4/10)**
>
> **Q10) Line 137: The statement about joint training vs. fine-tuning seems based on one specific example (COCO to Cityscapes). How does joint training on COCO and LVIS compare to fine-tuning on COCO?**
>
> As the reviewer noted, there is a distribution gap between the Cityscapes and COCO datasets, which may negatively impact the effectiveness of joint training. Using datasets with a smaller distribution gap, as suggested, could potentially yield different results. To explore this, we conducted experiments comparing joint training on COCO and ADE20K with fine-tuning on ADE20K alone. The results are presented in the table below and in Figure 2(b) of the revised manuscript.
>
> | **Method**              | **iter: 0** | **20k** | **40k** |
> |--------------------------|-------------|---------|---------|
> | Fine-tuning (no source)  | 23.5        | 46.3    | 48.2    |
> | Fine-tuning (with both)  | 23.5        | 43.0    | 44.9    |
> | From scratch (with both) | 0.0         | 43.3    | 45.7    |
>
> Our findings indicate that fine-tuning on a specific dataset consistently delivers greater performance improvements than joint training. Combined with the original experiments in the manuscript, this further emphasizes that joint training is less effective than fine-tuning, even when the distribution gap between datasets is relatively small. This experiment and analysis have been incorporated in Section 3.1.
>
> Note that, while we initially considered joint training on COCO and LVIS, LVIS was deemed unsuitable because it provides only instance annotations, whereas OVS models require panoptic annotations. Instead, we selected ADE20K, which shares similar characteristics with COCO and includes the necessary panoptic annotations.
>
> ---
>
> **Q11) Line 139: Why not use downsampling and upsampling to control that?**
>
> In joint training, the number of images in the training datasets can vary, leading to a class imbalance issue, which may hinder effective learning. Addressing class imbalance is known to be challenging [1]. Downsampling techniques can be applied to mitigate this issue; however, they risk losing important information during the reduction process. Similarly, oversampling techniques can distort the actual data distribution, potentially degrading generalization performance [2].
>
> Given the inherent challenges of addressing class imbalance in joint training, we believe this is a challenging problem that would benefit from dedicated research. Rather than attempting to solve this within the scope of our current work, we suggest that exploring effective strategies to manage class imbalance in multi-dataset training would be a valuable direction for future studies. We have included this discussion in Section 3.1.
>
> References:
>
> 1. Ghosh, Kushankur, et al. "The class imbalance problem in deep learning." *Machine Learning* 113.7 (2024): 4845-4901.
> 2. Tarawneh, Ahmad S., et al. "Stop oversampling for class imbalance learning: A review." *IEEE Access* 10 (2022): 47643-47660.
>
> ---
>
> **Q12) Line 136: For the first issue, why is it challenging? Is it hard to just consume some storage to save some data? Any big difference from storing means and covariance matrices for each dataset in this paper?**
>
> Data management is not limited to storage concerns but also involves challenges related to data usage rights, such as licensing. For example, if the usage rights of a dataset previously used for training expire, joint training becomes impossible. However, storing the means and variances of each dataset separately can mitigate these issues.
>
> We recognize that our previous explanation of source dataset management lacked sufficient detail. To address this, we have included the above example in Section 3.1.

---

> ### Author Response · Authors · 2024-11-20
> **Response to Reviewer mCxt (5/10)**
>
> **Q13) Line 157: What does PEFT mean?**
>
> PEFT (*Parameter-Efficient Fine-Tuning*) focuses on improving the performance of downstream tasks by fine-tuning only a subset of the model's parameters. Recently, PEFT has been widely adopted in the field of continual learning, where it is used to introduce dedicated model parameters for learning new data.
>
> We acknowledge that we did not provide an explanation of Parameter-Efficient Fine-Tuning (PEFT) earlier. To address this, we have added the above description to Section 2, *Related Work*, where PEFT is mentioned.
>
> ---
>
> **Q14) Figure 3(b): Could you clarify what “average inference time per sample” means? What exactly does “Number of seen datasets” represent—does it include both source datasets and newly added datasets? Additionally, why does the inference time vary based on the “Number of seen datasets”? Shouldn’t inference time depend only on the model, the testing sample, and the device used? Lastly, could you specify which datasets are referenced here?**
>
> Thank you for your questions about Figure 3(b). We would like to clarify the terms and concepts to ensure better understanding:
>
> 1. **"Average inference time per sample"** refers to the average amount of time required for a single sample to pass through the model during inference.
>
> 2. **"Number of seen datasets"** includes both the datasets the model was initially trained on (source datasets) and any newly added datasets. In this context:
>    - A value of 1 means the model was trained only on the COCO dataset.
>    - A value of 2 means the model was further trained on Cityscapes in addition to COCO.
>    - A value of 3 indicates additional training on ADE20K alongside COCO and Cityscapes.
>
> 3. **Why does inference time vary with the number of seen datasets?**
>    As the reviewer mentioned, inference time is generally determined by the model, the testing sample, and the device. However, in the continual learning context, inference time can also be influenced by the number of datasets involved. For instance:
>    - In methods like PEFT, where the number of introduced parameters increases linearly with the addition of datasets, the inference time also increases.
>    - Similarly, our method requires additional computations, such as applying the MVN distribution or interpolating weights between models, as the number of learned datasets increases.
>
> | **# Seen Datasets** | **PEFT (ms)**      | **Ours (ms)**        |
> |---------------------|------------------------|----------------------|
> | 1                   | 199.99 (+0.00%)       | 199.99 (+0.00%)     |
> | 2                   | +36.04 (+18.02%)      | +11.49 (+5.75%)     |
> | 3                   | +84.73 (+42.37%)      | +14.24 (+7.12%)     |
>
>    Specifically, for our method, the additional time incurred by these processes is negligible compared to the overall forward propagation time. When adding one more dataset after the model has already seen two datasets, the increase in inference time due to these additional computations accounts for only about **1.37%p of the total inference time**, as shown in the table above. On the other hand, for PEFT, the inference time increases by **24.4%p of the total inference time** with the addition of each dataset. This increase is non-negligible and highlights the advantage of our method, which maintains a much smaller increase in inference time.
>
> All these details have been incorporated into the caption of Figure 3(b) and the main text to provide a clearer explanation. Please share any further questions or concerns.
>
> ---
>
> **Q15) I think the problem definition is not open-vocabulary. For OV, the testing images and class is unknown, but in this paper, it allows sequential fine-tuning. So I think the paper is just a continual learning of segmentation.**
>
> As the reviewer mentioned, addressing the issue of sequential fine-tuning is indeed a key challenge of our paper. However, maintaining and improving performance on target datasets that were not used during training is also a significant challenge. In other words, our goal is to continually train the OVS model to expand its ability to predict across a broader data distribution while simultaneously preserving or enhancing its open-vocabulary capabilities. Therefore, we believe that the problem addressed in this paper aligns with the open-vocabulary domain.
>
> We hope this response has addressed the reviewer’s concern. If there are any additional questions or points that require further clarification, please let us know.

---

> ### Author Response · Authors · 2024-11-20
> **Response to Reviewer mCxt (6/10)**
>
> **Q16) For Algorithm 1, saving means and covariance matrices for each dataset may not be ideal. Why not do this for subgroups within datasets instead? The domain gap between two images across datasets may not always be larger than that within a single dataset. A more reasonable approach might be to cluster all seen data and apply the algorithm to each cluster.**
>
> As the reviewer pointed out, we agree that clustering all seen datasets and applying the algorithm to each cluster would be a more reasonable approach. This method could help accurately estimate the data distribution gap between samples. However, in our scenario, accessing all seen datasets is not feasible. As mentioned earlier, our task assumes that previously trained datasets are no longer accessible. Therefore, it is challenging to adopt the clustering-based approach suggested by the reviewer.
>
> Our proposed MVN distribution-based method is designed to handle both scenarios effectively:
>
> 1. **When the dataset gap is large:**
>    In cases where the distribution gap between datasets is large, the algorithm works as intended, assigning the sample to the most appropriate dataset distribution.
>
> 2. **When the dataset gap is small:**
>    Even when the distribution gap between datasets is smaller than the intra-dataset variation, the algorithm remains effective. This is because:
>    - **Highly similar distributions:** The datasets in question often share highly similar distributions. For instance, many datasets used in our experiments (e.g., COCO and ADE20K) exhibit overlapping characteristics in their data distribution. In such cases, even if the algorithm interpolates between these distributions, it effectively captures the shared structure without significant loss in performance.
>    - **Flexible interpolation:** The interpolation process allows for flexible handling of samples near the boundary of two distributions. This ensures that samples are not rigidly assigned to a single dataset but rather are evaluated in the context of both, leveraging their similarities to improve predictions.
>
> In essence, the similarity in dataset distributions, coupled with the interpolation mechanism, ensures robust performance even in cases where inter-dataset variation is minimal. We believe this makes the algorithm well-suited for practical scenarios where datasets may naturally exhibit overlapping or closely related distributions.
>
> We have revised Section 5.2 of the manuscript to include this discussion and to emphasize the versatility of our approach. Thank you for highlighting this point, and we hope this addresses your concern.
>
> ---
>
> **Q17) Overall, the MVN distribution seems similar to typical multi-domain adaptation methods, involving interpolation of source domains for target adaptation. Could Section 2 include a brief review of multi-domain adaptation literature?**
>
> Thank you for the suggestion. Our method shares similarities with multi-source domain adaptation (MSDA) methods while also having distinct differences. MSDA techniques align with our approach in that they utilize multiple models [1, 2] trained on diverse datasets. However, due to the following differences, MSDA techniques are not well-suited to our task:
>
> 1. **Simultaneous Access to Source Data**:
>    MSDA assumes that all source domain data is available simultaneously for training, whereas our method operates under a sequential learning setting where access to previously seen data is restricted.
>
> 2. **Focus on a Single Target Domain**:
>    MSDA techniques aim to optimize performance for a single target domain, which makes them unsuitable for open-vocabulary segmentation tasks where generalization performance is critical.
>
> 3. **Performance on Source Datasets**:
>    MSDA does not account for performance on the source datasets, making it unable to address the issue of catastrophic forgetting.
>
> 4. **Applicability of MSDA to Our Scenario**:
>    As mentioned above, MSDA requires simultaneous access to multiple source datasets for training, which is not possible in our scenario, where access to source domains is sequentially restricted. Additionally, the *"domain"* in MSDA typically refers to style or texture variations. This makes MSDA techniques unsuitable for scenarios like ours, where class labels change across datasets.
>
> Thus, while our approach shares certain similarities with MSDA, there are clear differences that set it apart. Comparing our task and method to MSDA techniques is meaningful, and we have therefore added a literature review of MSDA to Section 2.
>
> References:
>
> 1. Li, Yunsheng, et al. "Dynamic transfer for multi-source domain adaptation." *CVPR 2021*.
> 2. Song, Xiyu, et al. "P3-MSDA: Multi-source domain adaptation network for dynamic visual target detection." *Frontiers in Human Neuroscience* 15 (2021): 685173.

---

> ### Author Response · Authors · 2024-11-20
> **Response to Reviewer mCxt (7/10)**
>
> **Q18) For the experiments, I believe a crucial baseline is missing: joint training on all previous datasets combined with fine-tuning on the new datasets.**
>
> | **Method**       | **COCO (source)** | **Cityscapes (fine-tuning)** | **ADE20K (target)** |
> |-------------------|-------------------|-----------------------------|---------------------|
> | fc-CLIP          | 50.1              | 44.0                        | 23.5                |
> | + Fine-tuning    | -22.7             | +20.1                       | -10.3               |
> | + Joint Training | +0.6              | +17.9                       | +1.7                |
> | + Ours           | +0.3              | +20.2                       | +2.5                |
>
> | **Method**       | **COCO (source)** | **ADE20K (fine-tuning)** | **Cityscapes (target)** |
> |-------------------|-------------------|--------------------------|--------------------------|
> | fc-CLIP          | 50.1              | 23.5                     | 44.0                     |
> | + Fine-tuning    | -7.7              | +24.1                    | -3.0                     |
> | + Joint Training | +1.4              | +16.5                    | -1.2                     |
> | + Ours           | +1.7              | +23.8                    | -0.3                     |
>
> As suggested by the reviewer, we have added the results of joint training to Table 2. The results show that for the source dataset (COCO), performance is maintained or improved, and it achieves similar performance compared to our method. However, on the fine-tuning dataset, we observed that the performance improvement of joint training is relatively smaller compared to our method or fine-tuning alone. Additionally, for the target dataset, joint training performs worse than our method.
>
> | Method           | The order of fine-tuning | COCO (source) | ADE20K (fine-tuning 1) | Cityscapes (fine-tuning 2) |
> |-------------------|--------------------------|---------------|-------------------------|----------------------------|
> | fc-clip          | -                        | 50.1          | 23.5                   | 44.0                       |
> | + Fine-tuning    | ADE → City               | 20.8          | 15.4                   | 65.2                       |
> | + Fine-tuning    | City → ADE               | 39.3          | 48.3                   | 46.0                       |
> | + Joint Training | City, ADE                | 48.6          | 35.5                   | 60.5                       |
> | + Ours           | City, ADE                | **51.6**      | **47.0**               | **64.3**                   |
>
>
> | Method           | Source dataset         | The order of fine-tuning    | LVIS  | BDD100K | Mapillary | PC-59 | PC-459 | PAS-20 | PAS-21 | A-847 |
> |-------------------|------------------------|-----------------------------|-------|---------|-----------|-------|--------|--------|--------|-------|
> | fc-clip          | COCO                  | -                           | 20.5  | 19.0    | 26.0      | 53.0  | 16.9   | 93.1   | 80.2   | 13.8  |
> | + Fine-tuning    | COCO                  | Cityscapes → ADE20k         | 21.7  | 19.7    | 27.8      | 52.1  | 17.2   | 92.3   | 76.7   | 16.0  |
> | + Fine-tuning    | COCO                  | ADE20k → Cityscapes         | 10.4  | 21.3    | 24.2      | 45.9  | 13.5   | 87.4   | 70.7   | 11.5  |
> | + Joint Training | COCO, Cityscapes, ADE20k | -                           | 21.5  | 21.8    | 28.0      | 53.2  | 17.3   | 93.3   | **80.9** | 15.2  |
> | + Ours           | COCO                  | Cityscapes, ADE20k          | **23.1** | **22.6** | **29.1** | **54.9** | **17.9** | **93.6** | 80.7 | **16.3** |
>
> Additionally, we have included the results of joint training on the three datasets (COCO, Cityscapes, ADE20K) in Tables 3 and 4. As shown in Table 3, joint training achieves higher performance on the source dataset (COCO) compared to sequential fine-tuning but performs worse than our method across all three datasets. Furthermore, as demonstrated by experiments on eight different target datasets, joint training shows better generalizability than sequential fine-tuning but still underperforms compared to our method. These results and analyses have been added to Section 5.1.

---

> ### Author Response · Authors · 2024-11-20
> **Response to Reviewer mCxt (8/10)**
>
> **Q19) Why not compare to other OVS methods like ODISE, OpenSeeD, Open-vocabulary SAM, etc? The setting is easy, just use all previous and fine-tuning datasets to train their models and evaluate on these eight unseen datasets. Otherwise, the results shown like in Tables 3, 4 make it hard for the reader to believe the proposed method works.**
>
> Thank you for the valuable feedback. Our method is designed to be easily adaptable to other techniques, so we focused on evaluating how much it improves the performance of existing methods rather than directly comparing it with other OVS methods. However, we agree with the reviewer that comparing our approach to other OVS methods is essential to demonstrate the reliability of our performance metrics.
>
> To address this, we conducted a comparison between our method and OpenSeeD, which uses COCO and Object365 as source datasets. Object365 is a large-scale dataset containing 2 million images and 365 classes, and [1] reported that it improved OpenSeeD's generalization performance. The results of this comparison are provided in the table below.
>
> | **Method**        | **Source Dataset**          | **Order of Fine-tuning** | **LVIS** | **BDD100K** | **Mapillary** | **PC-59** | **PC-459** | **PAS-20** | **PAS-21** | **A-847** |
> |--------------------|-----------------------------|---------------------------|----------|-------------|---------------|-----------|------------|------------|------------|-----------|
> | OpenSeeD          | COCO, Object365             | -                         | 14.4     | 10.7        | 15.0          | 47.7      | 11.0       | 87.2       | 33.5       | 5.3       |
> | fc-CLIP           | COCO                        | -                         | 20.5     | 19.0        | 26.0          | 53.0      | 16.9       | 93.1       | 80.2       | 13.8      |
> | + Joint Training  | COCO, Cityscapes, ADE20K    | -                         | 21.5     | 21.8        | 28.0          | 53.2      | 17.3       | 93.3       | **80.9**   | 15.2      |
> | + Fine-tuning     | COCO                        | Cityscapes → ADE20K       | 21.7     | 19.7        | 27.8          | 52.1      | 17.2       | 92.3       | 76.7       | 16.0      |
> | + Fine-tuning     | COCO                        | ADE20K → Cityscapes       | 10.4     | 21.3        | 24.2          | 45.9      | 13.5       | 87.4       | 70.7       | 11.5      |
> | + Ours            | COCO                        | Cityscapes, ADE20K        | **23.1** | **22.6**    | **29.1**      | **54.9**  | **17.9**   | **93.6**   | 80.7       | **16.3**  |
>
> As shown in the table, we observe that OpenSeeD performs worse than fc-CLIP, which is trained solely on COCO, across the eight target datasets. Furthermore, our method shows significantly higher performance than OpenSeeD across all datasets. We have included these results and their analysis in Section 5.1.
>
> As a note, we were unable to conduct joint training experiments on COCO, ADE20K, and Cityscapes using ODISE, OpenSeeD, and Open-vocabulary SAM due to limited resources. Specifically, training ODISE and OpenSeeD on COCO alone requires more than 6 days with 32 V100 GPUs, and Open-vocabulary SAM demands 8 A100 GPUs. Training on all the datasets mentioned would likely take longer. Although we could not fully complete the requested experiments, we hope that the provided OpenSeeD results address your concerns. If you have any additional suggestions, please feel free to share them.
>
> Reference:
>
> 1. Zhang, Hao, et al. "OpenSeeD: Open-vocabulary semantic segmentation with pre-trained vision-language models." *CVPR 2023*.

---

> ### Author Response · Authors · 2024-11-20
> **Response to Reviewer mCxt (9/10)**
>
> **Q20) Another big problem is the experiments need cross validation, not just usually COCO as the previous training set. In the real world, we could start with ADE, we could start with Cityscapes, or we could also start with a subset like 10K training samples of COCO. In these cases, does the proposed method still work well?**
>
> | **Method**       | **ADE20K (source)** | **COCO (fine-tuning)** | **Cityscapes (target)** |
> |-------------------|---------------------|------------------------|--------------------------|
> | fc-CLIP          | 48.1                | 42.3                   | 40.9                     |
> | + Fine-tuning    | -18.5               | +10.4                  | +3.3                     |
> | + Ours           | -1.3                | +9.3                   | +5.2                     |
>
> | **Method**       | **ADE20K (source)** | **Cityscapes (fine-tuning)** | **COCO (target)** |
> |-------------------|---------------------|-----------------------------|-------------------|
> | fc-CLIP          | 48.1                | 40.9                        | 42.3              |
> | + Fine-tuning    | -18.5               | +21.4                       | -11.5             |
> | + Ours           | +0.0                | +19.5                       | +0.0              |
>
> Based on the reviewer’s suggestion, we conducted additional experiments to analyze whether our method performs well when a dataset other than COCO (e.g., ADE20K) is used as the source dataset. The results, which have been added to the tables above, demonstrate that our method maintains performance on the source dataset (ADE20K) while improving performance on the fine-tuning dataset. Furthermore, it enhances performance on target datasets that were not used during training.
>
> We acknowledge that we had not previously validated how performance changes when the source dataset is different, and we have included these experimental results and analyses in Section 5.1. Thank you for your valuable question.

---

> ### Author Response · Authors · 2024-11-20
> **Response to Reviewer mCxt (10/10)**
>
> We hope that we have addressed your concerns regarding our work. Additionally, we have revised the manuscript in accordance with your comments and kindly request your further feedback at your earliest convenience.

---

> ### Author Response · Authors · 2024-11-27
> **Follow-Up on Our Response to Your Feedback**
>
> Dear Reviewer mCxt,
>
> Thank you once again for your thoughtful and detailed review of our paper. We have carefully addressed all of your feedback and incorporated the suggested changes into the manuscript.
>
> If there are any additional questions, concerns, or suggestions, we would be more than happy to address them promptly. Please do not hesitate to share any further input or clarification you may require. We remain committed to ensuring that our responses align with your expectations and hope they have satisfactorily resolved the points you raised.
>
> Thank you for your time and consideration, and we look forward to hearing from you at your earliest convenience.
>
> Sincerely,
> The Authors

---

> ### Author Response · Authors · 2024-11-28
> **Response to Additional Feedback from Reviewer mCxt (3/3)**
>
> **Q3) I just re-read the revised problem definition section, lines 224-234. It is still unclear. Line 230, what is exactly ‘seen’ datasets, source datasets + a bunch of fine tuned datasets? If so, this conflicts with the statement in the paper and rebuttal where for example, it states that in the i-th fine-tuning stage, the model has access only to the current dataset, D^i_ft. Why can't other D^i_ft sometimes be accessed but during evaluation, it is suddenly accessible.**
>
>
> First, regarding the definition of "seen" datasets, as noted by the reviewer, the term "seen" datasets in our paper refers to both the source dataset and the fine-tuning datasets used during the fine-tuning process. We recognize that this term may lack clarity and have revised it to "source and fine-tuning datasets" for greater precision.
>
> Next, we explain how datasets are accessed during training and evaluation. In our study, during the training phase, the model can only access the training set of the current fine-tuning dataset. In other words, during the fine-tuning stage, the model does not have access to previously used fine-tuning datasets or the source dataset. On the other hand, for the purpose of final performance evaluation, the test sets of previously used fine-tuning datasets are accessed. This approach aligns with common practices in machine learning research. For example:
>
> - **Unsupervised Domain Adaptation (UDA)**: The target dataset is not accessible during training but is used during evaluation to measure performance.
> - **Federated Learning**: The central server cannot directly access the target dataset during training, but it is utilized during evaluation to validate model performance.
> - **Few-shot Classification**: The target dataset is inaccessible during training but is used during evaluation to assess the model's performance.
>
> Note that, we clarify that these previously fine-tuning datasets cannot be accessed for validation purposes.
>
> To address the reviewer's concern and enhance clarity, we have revised the entire problem definition section. The updated content is as follows:
>
> > OVS models often struggle with desired unseen data distributions, limiting their applicability in real-world scenarios where new objects or classes frequently emerge. For instance, consider a scenario where a user deploys an OVS model in a driving scene. Pre-trained OVS models, without fine-tuning, perform poorly because they have not adapted to the driving scene’s data distribution. On the other hand, fine-tuning the model on such data can compromise its open-vocabulary capabilities, restricting it to recognizing only objects and classes typical of the driving scene. This study addresses this issue by proposing a method that sequentially fine-tunes the model, extending its data distribution coverage while preserving its open-vocabulary properties.
>
> > More formally, we define our objective in detail as follows: The OVS model is first trained on the source dataset D^{train}\_{pr} and then fine-tuned sequentially on specific datasets {D^{train}\_{ft,1}, D^{train}\_{ft,2}, . . . }. Each dataset D contains images X\_{img} and class labels X\_{text}. At the i-th fine-tuning stage, the model only has access to the current training set D^{train}\_{ft,i} of the fine-tuning dataset. For evaluation, the model is evaluated on the test sets of source and fine-tuning datasets {D^{test}\_{pr}, D^test\_{ft,1}, . . . , D^{test}\_{ft,i}}, and target datasets {D\_{target,1}, D\_{target,2}, . . . }. Note that the model has never encountered the target datasets during training.
>
> We hope these clarifications address your concerns, and we appreciate your insightful feedback.
>
>
>
> **Q4) And I don’t understand this “The main challenge is to improve performance on the new dataset D^i_ft? D^i_ft is a new dataset? Shouldn’t it be the fine-tuned datasets as stated. Has it been used during training?**
>
> To clarify, \( D^i_{ft} \) refers to the fine-tuning dataset used to train the model during the fine-tuning stage. We apologize for any confusion caused by this. As outlined in our response to Q3, we have revised this paragraph to eliminate the ambiguity. We hope this revision resolves any remaining uncertainties. Thank you once again for your constructive feedback.

---

> ### Author Response · Authors · 2024-12-01
> **Kind Follow-Up on Discussion Period Feedback**
>
> Dear Reviewer,
>
> With only two days left in the discussion period, we kindly follow up on our earlier reminders regarding your feedback. We have carefully worked to address the questions and concerns you raised through our responses and revisions, and we believe that we have resolved most, if not all, of the issues you highlighted.
>
> If our efforts and revisions meet your expectations, we would greatly value your acknowledgment. However, if you have any additional concerns, we are more than willing to address them promptly before the discussion period concludes.
>
> Thank you for your time and thoughtful consideration.
>
> Sincerely,
> The Authors

---

### Author Response · Authors · 2024-11-26
**General Response**

We sincerely thank **Reviewers mCxt, bfmq, pQF1, and xh2L** for their detailed and thoughtful reviews, as well as their valuable and insightful feedback. We appreciate the positive comments from all reviews:

- **Reviewer mCxt**
  1. Easy-to-understand and implementable method.
  2. Comprehensive experiments and ablation study on fc-clip.

- **Reviewer bfmq**
  1. Adaptability to new domains without retraining or prior dataset access.
  2. Mitigation of catastrophic forgetting while maintaining previous performance.
  3. Parameter-efficient integration with existing architectures.

- **Reviewer pQF1**
  1. Proposes a clear and intuitive approach.
  2. Validated through experiments on diverse datasets and tasks.

- **Reviewer xh2L**
  1. Addresses the poor performance of OVS models on unseen domains.
  2. Demonstrates robustness and generalizability across diverse datasets.

We have thoroughly revised the paper to address all the reviewers' comments. All newly added or modified content has been **highlighted in blue** throughout the revised manuscript. Below, we provide a summary of the changes made:

---

### **Reviewer mCxt**

1. Replaced the term "domain" with "data distribution" throughout the revised manuscript (**Q1**).
2. Updated Table 1 to ensure a fair comparison in Section 1 (**Q2**).
3. Added references to clarify Lines 58–59 (**Q4**).
4. Provided assumptions regarding the new dataset in Lines 62–63 for clarification (**Q5**).
5. Replaced "previous" with "source" and "unseen" with "target" throughout the revised manuscript (**Q6**).
6. Included OVS model details in the caption of Figure 1 (**Q7**).
7. Revised the expression “fail to generalize to new domains” in Lines 96–97 (**Q8**).
8. Expanded the explanation of batches with balanced proportions of samples in Section 3.2 (**Q9**).
9. Added experiments and analysis of joint training with different datasets in Section 3.1 (**Q10**).
10. Discussed strategies for addressing class imbalance in multi-dataset training in Section 3.1 (**Q11**).
11. Provided examples of data management challenges in Section 3.1 (**Q12**).
12. Added an explanation of Parameter-Efficient Fine-Tuning (PEFT) in Section 2 (**Q13**).
13. Clarified the terms and underlying concepts of Figure 3(b) in its caption (**Q14**).
14. Explained how the dataset gap is managed when using the MVN distribution in Section 5.2 (**Q16**).
15. Added a literature review of MSDA in Section 2 (**Q17**).
16. Included experimental results and analyses of joint training in Section 5.1 and updated Tables 2, 4, and 5 (**Q18**).
17. Provided experimental results and analyses of OpenSeeD in Section 5.1 and Table 5 (**Q19**).
18. Included experimental results and analyses when using ADE20k as the source dataset in Section 5.1 and Table 3 (**Q20**).

---

### **Reviewer bfmq**

1. Analyzed the computational resources required for our method in Section 5.3 (**Q1**; **Q2 by Reviewer xh2L**).
2. Clarified the rationale for selecting diverse metrics in Tables 5 and 6 in Appendix B (**Q2**).
3. Discussed strategies for managing the computational cost of our method in Section 6 (**Q3**).

---

### **Reviewer pQF1**

1. Clarified our contributions to the OVS task, including the evaluation protocol, in Section 1 (**Q1**).
2. Defined the objectives of our tasks more clearly in Section 3 (**Q2**).
3. Included experimental results and analyses for truly unseen classes in Section 5.2 (**Q3**).
4. Added experimental results for unseen datasets to the ablation studies in Section 5.2 (**Q4**).
5. Discussed the applicability of our method to traditional continual learning tasks in Appendix D (**Q5**).
6. Provided an analysis of the storage demand in Section 5.3 (**Q6**).

---

### **Reviewer xh2L**

1. Added experimental results and analyses for testing across a wider range of domains in Section 5.2 (**Q1**).
2. Included a detailed analysis of hyperparameter sensitivity in Appendix D (**Q3**).

---

Detailed responses to other comments are left in individual comments.

---

### Meta-Review · Area_Chair_yBkT · 2024-12-17

**Metareview:**

The paper proposes to combine continual learning and open vocabulary image segmentation to improve the generalizability of the model across different domains. The paper received three negative ratings, 3, 5, 5, and one positive ratings 6. The reviewers find that this paper is clear and easy to understand, its performance is good across datasets. However, some critical issues are pointed out in the reviews: unconvinced experiments that lacks persuasiveness, limited diversity of domains in experiments, the robustness and effectiveness of the method on real-world filed remain doubtful, the proposed method may not have a significant impact on the open vocabulary task. The rebuttal does not fully address these concerns. After rebuttal, all reviewers keep the initial rating, Reviewer xh2L keep borderline score 6 after reading the authors' rebuttal. Given the concerns raised by reviewers, the AC recommends rejection. The authors are encouraged to consider the reviewers' comments when revising the paper for submission elsewhere.

**Additional Comments On Reviewer Discussion:**

All the reviewers keep the initial ratings. Reviewer xh2L gave borderline accept and raised concerns about testing dataset, the usage of MVNs, and detailed analysis of hyperparameters, after rebuttal, Reviewer xh2L keep the initial rating. All the reviewers raised concerns regarding unconvinced experiments. There is a consensus that this paper is not ready for ICLR.

---

### Decision · Program_Chairs · 2025-01-22

Reject